# Sequence heterochrony led to a gain of functionality in an immature stage of the central complex: A fly–beetle insight

**Max S. Farnworth**[1,2], **Kolja N. Eckermann**[2,3], **Gregor Bucher**[1]*

**1** Department of Evolutionary Developmental Genetics, Johann-Friedrich-Blumenbach Institute, GZMB, University of Göttingen, Göttingen, Germany, **2** Göttingen Graduate Center for Neurosciences, Biophysics and Molecular Biosciences (GGNB), Göttingen, Germany, **3** Department of Developmental Biology, Johann-Friedrich-Blumenbach Institute, GZMB, University of Göttingen, Göttingen, Germany

* gbucher1@uni-goettingen.de

**Data Availability Statement:** The stacks of all pictures shown in the paper are available in .tif and .avi formats from the figshare database

## Abstract

Animal behavior is guided by the brain. Therefore, adaptations of brain structure and function are essential for animal survival, and each species differs in such adaptations. The brain of one individual may even differ between life stages, for instance, as adaptation to the divergent needs of larval and adult life of holometabolous insects. All such differences emerge during development, but the cellular mechanisms behind the diversification of brains between taxa and life stages remain enigmatic. In this study, we investigated holometabolous insects in which larvae differ dramatically from the adult in both behavior and morphology. As a consequence, the central complex, mainly responsible for spatial orientation, is conserved between species at the adult stage but differs between larvae and adults of one species as well as between larvae of different taxa. We used genome editing and established transgenic lines to visualize cells expressing the conserved transcription factor *retinal homeobox*, thereby marking homologous *genetic neural lineages* in both the fly *Drosophila melanogaster* and the beetle *Tribolium castaneum*. This approach allowed us for the first time to compare the development of homologous neural cells between taxa from embryo to the adult. We found complex heterochronic changes including shifts of developmental events between embryonic and pupal stages. Further, we provide, to our knowledge, the first example of *sequence heterochrony* in brain development, where certain developmental steps changed their position within the ontogenetic progression. We show that through this *sequence heterochrony*, an immature developmental stage of the central complex gains functionality in *Tribolium* larvae.

## Introduction

The brain is among the most complex organs of an animal, in which sensory inputs and internal states are processed to guide behavior. Hence, evolutionary modifications of brain structure and function in response to different life strategies are paramount for a species'

([https://figshare.com/projects/Additional_data_for_Farnworth_et_al_/64799](https://figshare.com/projects/Additional_data_for_Farnworth_et_al_/64799)).

**Funding:** Final work of this project was supported by Deutsche Forschungsgemeinschaft grant BU1443/17-1 to GB ([www.dfg.de](www.dfg.de)) MSF and KNE were supported by the Göttingen Graduate Center for Molecular Biosciences, Neurosciences and Biophysics (GGNB) ([http://www.uni-goettingen.de/de/56640.html](http://www.uni-goettingen.de/de/56640.html)) The funders had no role in study design, data collection and analysis, decision to publish, or preparation of the manuscript.

**Competing interests:** The authors have declared that no competing interests exist.

**Abbreviations:** ac, anterior commissure; AL, antennal lobe; CA, calyx; CB, central body; CM, centro-medial; CX, central complex; D, dorsal; dlrFB, dorso-lateral root of the FB; DM, dorso-median; EB, ellipsoid body, lower unit of CB; EGFP, enhanced green fluorescent protein; FB, fan-shaped body, upper unit of CB; GFP, green fluorescent protein; GNG, gnathal ganglia; L1, first instar larva; lv, larval; lvCB, larval CB; MEF, medial equatorial fascicle; mrFB, medial root of the FB; n-, neuraxis referring; n-dorsal, neuraxis dorsal; NO, noduli; NS, neural stage; P2A, viral peptide inducing ribosome skipping; PB, protocerebral bridge; pc, posterior commissure; pr, primordial; prFB, primordial FB; RNAi, RNA interference; *rx*, retinal homebox gene; Rx, Retinal homeobox protein; SME, smooth manifold extraction; V, ventral.

adaptation, and indeed, the morphology of brains is highly diverse [1,2]. However, it is unclear how brains can evolve, despite the fact that the complex neural interconnectivity is essential for their function and hence, likely to restrict evolvability.

Any divergence in adult brain morphology traces back to modifications of the developmental program. One of the developmental mechanisms for brain diversification is heterochrony, i.e., the change in relative timing of a developmental process in one taxon compared with another [3–5]. For instance, heterochronic extension of the growth phase of a postnatal infant human brain compared with other primates leads to a relative increase of final brain size [3,6]. An intriguing heterochronic divergence in brain morphology was found with respect to the central complex (CX) of insects. In hemimetabolous insects, all brain neuropils develop during embryogenesis and already the hatchling has an adult-like CX [7,8]. By contrast, in holometabolous insects, the CX forms only partly during embryogenesis and is completed later during metamorphosis. In the tenebrionid beetles *Tenebrio molitor* and *T. castaneum*, for instance, the larval central body (lvCB) appears to consist of only the upper division, the fan-shaped body (FB). The lower division, the ellipsoid body (EB), was proposed to develop later during pupal stages [9–11]. In *Drosophila*, no functional CX neuropils are detectable in the first instar larva at all. At that stage, the CX primordium consist only of commissural tracts lacking neuropil morphology and synapses [12,13]. Only during late larval stages and metamorphosis, the CX matures into the adult form [13,14]. Intriguingly, the development at least of the FB appears to be quite similar between the hemimetabolan desert locust *Schistocerca gregaria* and the fly *D. melanogaster*, although they occur at different live stages [15]. This heterochronic divergence of the FB is thought to correlate with the development of walking legs, whereas the presence of the EB may be linked to the formation of complex eyes [1,8,16,17].

The CX integrates multisensory information and acts predominantly as spatial orientation and locomotor control center [16,17]. In adult insects, the CX structure is highly conserved, consisting of a set of midline-spanning neuropils, the protocerebral bridge (PB), the central body (CB) consisting of an upper (CBU or FB) and lower division (CBL or EB) and the noduli (NO) with stereotypical patterns of innervation (Fig 1A) [1,16,18–20]. The CX consists mainly of columnar and tangential neurons [16,19]. Tangential neurons connect other brain areas with one CX neuropil (Fig 1A) [16,21–23]. The focus of this work lies on columnar neurons, which connect the different neuropils of the CX with each other by projecting as 4 prominent tracts (called the WXYZ tracts) from the PB into FB, EB, NO, and other brain structures (Fig 1A) [15,19,24–27]. These neurons are required for the formation of the typical columnar architecture of the CB and PB [15,16,24,28]. Work in *S. gregaria* and *D. melanogaster* provided insights into how the complex innervation architecture develops and specifically, how fascicle switching leads to X-shaped crossings of neurites called decussations, which represent the cellular substrate of the columnar architecture of the CB [7,12,13,15,29–31]. An overview on the relevant developmental processes is summarized in Fig 1B.

The low number of neural cells in insect brains compared with vertebrates, the conservation of neural lineages building up the brain, and their experimental accessibility makes insects an excellent choice to study the mechanisms of brain diversification during development. Actually, recent technical advances have opened the possibility to study the genetic and cellular basis of brain development not only in the classic model organism *D. melanogaster* but also in other insects in order to reveal conserved and divergent aspects. The red flour beetle *T. castaneum* is spearheading comparative functional genetic work in neurogenesis because of its well-developed genetic toolkit and advances in neurobiological methods [9,37–47]. Recently, we suggested to compare homologous cells in different taxa by marking what we called *genetic neural lineages*, i.e., cells that express the same conserved transcription factor [9]. Essentially, this approach assumes that transcription factors with conserved expression in the

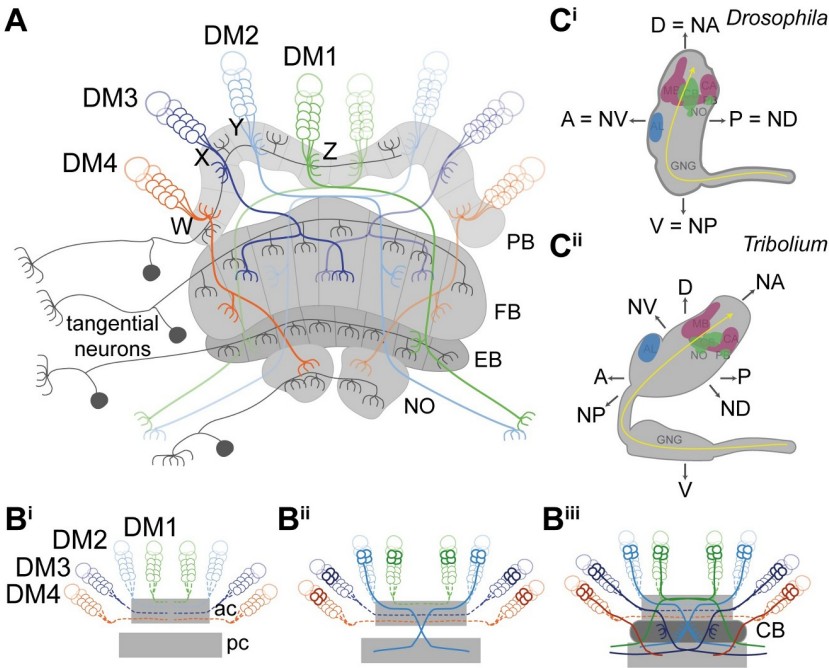

**Fig 1. Structure and development of the central complex, and relationship of neuraxis to body axes. (A)** Tangential neurons (dark gray) connect neuropils of the central complex with other areas. Columnar neurons (colored) connect the different neuropils of the central complex with each other. Nearly all columnar neurons derive from 4 type II neuroblasts, DM1-4 (green, light blue, dark blue, orange) that project through WXYZ tracts. **(B)** Central complex development starts with the neurons of the DM1-4 lineage (alternative names in *Drosophila*: DPMm1, DPMpm1, DPMpm2, CM4 or in *Schistocerca*: ZYXW) projecting into an ac (hatched lines in B$^i$) where they cross the midline and build up a stack of parallel fibers. Later-born neurons (solid lines in B$^{ii}$) undergo fascicle switching, i.e., they leave the ac-fascicle at stereotypical locations and re-enter a fascicle of a pc, forming X-shaped crossings with neurons from the contralateral side (called decussations) (B$^{ii}$). Decussations occur at different positions subdividing the future central body into columns (B$^{iii}$). PB omitted for simplicity; based on [15,22,32,33]. **(C)** The *Drosophila* (C$^i$) and *Tribolium* (C$^{ii}$) brains differ in their orientation within the head (lateral views). Although the *Drosophila* brain is oriented perpendicular to the ventral nerve cord, the *Tribolium* brain is tilted backwards. This leads to discrepancies when using the body axis as reference. For instance, the AL is anterior in *Drosophila*, whereas it is more dorsal in *Tribolium*. Similarly, the PB is posterior in *Drosophila* but rather ventral in *Tribolium*. To facilitate cross-species comparisons, we use the classic neuraxis nomenclature as suggested by [34]. In this system, the ALs are n-ventral (NV) and the PB n-dorsal (DV) in both species. Shapes of brains are based on v2.virtualflybrain.org/ and data from this study, whereas the shape of the *Tribolium* GNG is from [35]. Information about cell innervation in A was taken from [21,25,36]. A, anterior; ac, anterior commissure; AL, antennal lobe; CA, calyx; CB, central body; D, dorsal, DM, dorso-median; FB/EB, upper and lower division of the CB, respectively; GNG, gnathal ganglia; MB, mushroom body (excluding CA); n, neuraxis-referring; NO, noduli; NV, n-ventral; P, posterior; PB, protocerebral bridge; pc, posterior commissure; V, ventral.

neuroectoderm and the brains of most bilateria are likely to mark homologous cells in closely related taxa throughout development. It should be noted, however, that the actual identity of a given neuroblast lineage is not determined by a single transcription factor but by a cocktail of several factors [48]. *Genetic neural lineages* can be labeled either by classic enhancer trapping or a targeted genome editing approach, both available in *Tribolium* [39,41,47].

In this study, we investigated the cellular bases of heterochronic CX development by marking the *retinal homeobox (rx) genetic neural lineage* in both the red flour beetle *T. castaneum* and the vinegar fly *D. melanogaster* with antibodies and transgenic lines. We confirm that homologous cells are marked and subsequently scrutinize their embryonic and postembryonic development. We found a complex pattern of heterochrony underlying differentiation between larval and adult brains, including the shift of certain developmental events between

life stages. Intriguingly, we found that not only the timing but also the order of developmental steps was changed, representing a case of *sequence heterochrony*, which, to our knowledge, had not been observed in brain development before. As consequence, the larval CB of *Tribolium* is not a fully developed FB but represents an immature developmental stage, which gained functionality precociously. Apparently, CX functionality does not require the full connectivity as observed in adult brains.

## Results

### Marking the *rx* genetic neural lineage in 2 species

To compare CX development between 2 species, we wanted to mark a subset of homologous neurons that contribute to the CX. For this purpose, we decided to use the *rx genetic neural lineage* for 3 reasons: First, *rx* is one of the genes that is expressed almost exclusively in the anterior brain in bilaterians, indicating a highly conserved function in many animals [49–57]. Second, we had found projections into the CX in a *Tribolium rx* enhancer trap line and a small subset of CX projections in *Drosophila rx* VT-GAL4 lines (VDRC, # 220018, # 220016, discarded) [58,59]. Third, CX phenotypes were observed in both *Drosophila* and *Tribolium* in a *Dm-rx* mutant and *Tc-rx* RNA interference (RNAi) knockdown, respectively, indicating an essential role in CX development [51,60].

To mark *rx genetic neural lineages*, we first generated and validated an antibody binding the *Tribolium* Rx protein (Tc-Rx, TC009911) (S1 Fig) and used an available *Drosophila* Rx (Dm-Rx, CG10052) antibody [51]. Next, we tested an enhancer trap in the *Tc-rx* locus (E01101; *Tc-rx*-EGFP line) [47] and confirmed coexpression of EGFP (enhanced green fluorescent protein) with Tc-Rx (S2 Fig). The enhancer trap marked a subset of Tc-Rx-positive cells in the region of the CX depending on the stage (77.39% in first instar larval [L1], 50.23% in the prepupa, 9.71% in the adult; S2 Fig). Importantly, all EGFP-positive cells were Tc-Rx-positive as well (S2 Fig). For *Drosophila*, we generated an imaging line using CRISPR/Cas9-mediated homology-directed repair (S3 Fig). We replaced the stop codon of the endogenous *rx* locus with a P2A peptide sequence (this viral peptide induces ribosomal skipping during translation such that two separate proteins are formed from one messenger RNA) followed by an EGFP coding sequence [39,61,62]. The resulting bicistronic mRNA led to translation of nonfused Dm-Rx and EGFP proteins (*Dm-rx*-EGFP; S3 Fig), and we found complete coexpression of Dm-Rx and EGFP. Based on both antibodies and transgenic lines, we tested the labeled cells for homology.

### Similar location of *rx*-positive neural cell groups in both species

To get an overview on the conservation of Rx expression between *Drosophila* and *Tribolium*, we first compared the location of Rx-positive cells by using antibody stainings in adult brains and embryos. Note that the axes of the brain relative to the body axes are not conserved in insects. Therefore, we describe the location according to the "neuraxis" for both species (see explanation in Fig 1C). We found 4 major domains of Rx-positive cells (I-IV) located in similar regions in both species and similar expression in the embryo (Fig 2; see stacks and videos of all projections shown in this paper on https://figshare.com/projects/Additional_data_for_Farnworth_et_al_/64799). Additionally, we mapped the labeled cell groups to the locations of known *Drosophila* neural lineages [63,64] (S4 Fig, S1 Table, and S1 Text).

Rx-positive cell groups likely belonged to 11 neural lineages projecting into many regions of the brain including CX, mushroom bodies, and other structures. Four lineages (DM1-4) were prominently marked in the imaging lines of both species. Because these lineages are

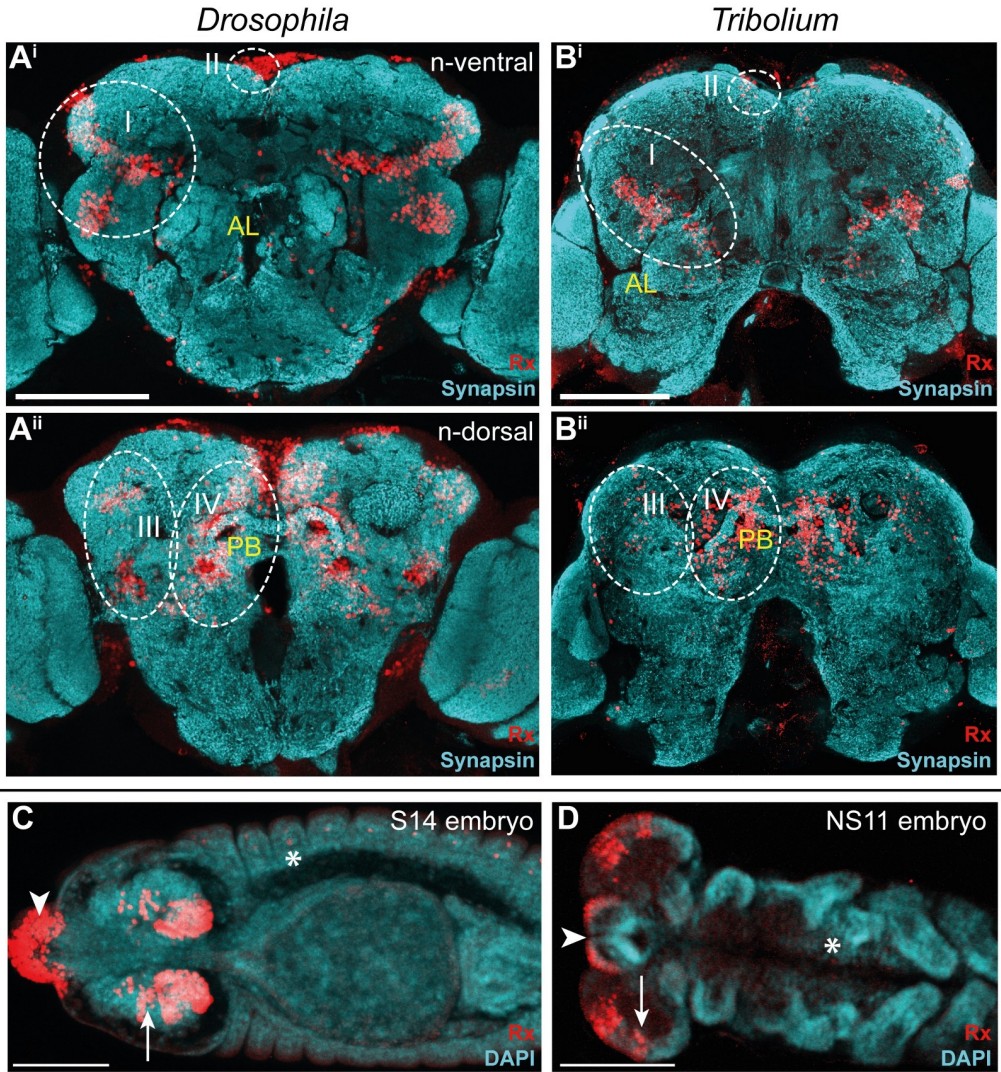

**Fig 2. Rx expression is conserved in *Drosophila* and *Tribolium* adult brains and embryos. (A-B)** Immunostainings revealed 4 domains of Rx-positive cells (I-IV, dotted white lines) with similar shape and position within the brain. Shown are n-ventral (i) and n-dorsal views (ii). **(C-D)** In *Drosophila* (S14) and *Tribolium* (NS11) embryos Rx was expressed in the labrum (arrowhead) and in similar regions of the lateral head neuroectoderm (arrows). In addition, single cells of the peripheral nervous system and ventral nerve cord were labeled in each segment (asterisk; S1 Fig). Note that the head lobes of *Tribolium* embryos are shown as flat preparations, whereas the *Drosophila* head was imaged within the egg. According to the *bend and zipper model* of head morphogenesis, the expression patterns are very similar [65]. Abbreviations like in Fig 1. Scale bars represent 100 μm. Original data: https://figshare.com/articles/Fig2_FigS4/11339795. AL, antennal lobes; CB, central body; n, neuraxis-referring; Rx, retinal homeobox protein.

known to contribute to the CX, we subsequently focused on the comparison of Rx-positive cell clusters of DM1-4.

## CX Rx-positive cell clusters are homologous between *Drosophila* and *Tribolium*

To corroborate the homology of Rx-positive DM1-4 neurons, we examined the location and projection pattern of these cell clusters in detail. We indeed found similar cell body locations

around the PB (Fig 3A and 3B) and similar projection patterns into the FB (Fig 3C and 3D), EB, and NO (Fig 3E and 3F) in both species. The similarity relative to CX neuropils was visualized in 3D reconstructions (Fig 3G and 3H, see videos on Figshare) and allowed us to define homologous cell clusters. Given the lack of a detailed map and homology assessments for the *Tribolium* brain, we assigned the fiber bundles medial equatorial fascicle (MEF), dorso-lateral root of the FB (dlrFB), and medial root of the FB (mrFB, see e.g., [12]) based on their similarity to the *Drosophila* brain and the novel lineage information gained in this study (S4 Fig, S1 Text).

Our classification of these Rx expressing cell clusters to lineages DM1-4 was supported in *Drosophila* by Rx immunostainings in the R45F08-GAL4 line, a *pointed* GAL4 enhancer construct that was suggested to label a large subset of neurons of the DM1-3 and 6 lineages [13]. Moreover, we crossed the *Dm-rx*-EGFP line to the R45F08-GAL4 line and found approximately 90% overlap (S5A and S5B Fig). In addition, a substantial part of the midline projections overlapped between both transgenic lines (S5C Fig).

Note that the *Dm-rx*-EGFP line marked all Dm-Rx-positive cells, whereas the *Tc-rx*-EGFP line marked only a subset of Rx-positive cell bodies (S2 Fig versus S3 Fig). This resulted in more prominently marked tracts in *Drosophila* compared with *Tribolium*. However, the *Tribolium* DM4 Rx expressing group showed a very high EGFP expression, such that the respective projections into the FB, NO, and EB as well as the connections to the lateral accessory lobes appeared much stronger than in *Drosophila* (Fig 3B, 3D and 3F[i]). This divergence of intensity was likely a particularity of the *Tribolium* enhancer trap.

In summary, we assume homology of the *rx*-positive cells of the DM1-4 lineages of *Drosophila* and *Tribolium* based on the shared expression of a highly conserved brain regulator and the specific similarity of cell body location of the DM1-4 lineages relative to the PB and their similar projection patterns in adult brains. Note that *rx* is expressed in most but probably not all cells of the DM1-4 lineages and in addition is expressed in cells contributing to other brain regions like the mushroom bodies, which were not examined here. The DM1-4 lineages are key components of the CX, providing nearly all columnar neurons [7,12,15]. Therefore, the *rx genetic neural lineage* is an excellent marker to compare CX development between fly and beetle.

## Divergent CX structures in the L1 larva of *Drosophila* and *Tribolium*

Next, we compared CX structures in the first instar larval (L1) brain of both species, since the strongest divergence between *Drosophila* and *Tribolium* seemed to occur at the larval stage. Here, tenebrionid beetle larvae have a partial CX neuropil already at the larval stage [9,11], whereas in *Drosophila* L1 larvae CX neuropil is missing [12]. We used synapsin and acetylated α-tubulin staining [67] to reveal functionality and underlying tract architecture, respectively (Fig 4).

As previously described [12], we found no functional (i.e., synapsin-positive) CX neuropil in *Drosophila* L1 (neither PB, CB, nor NO; Fig 4E[ii] and 4G[ii]). In *Tribolium*, in contrast, we observed a PB, which in synapsin stainings was nonfused (Fig 4F[ii]). Further, we found an lvCB, which showed no morphological sign of subdivision into upper or lower division (Fig 4H[ii]). Neither neuropil displayed an overt columnar structure in anti-synapsin or anti-GFP stainings (Fig 4F[ii] and 4H). Hence, the lvCB appeared as a simple bar-shaped neuropil as described before [9,11].

The analysis of *rx* expressing DM1-4 cells in *Drosophila* revealed that the spatial arrangement of marked cell bodies and their projections in the L1 differed from the adult (Fig 4E[i]—compare with Fig 3A). The cell clusters differed both in their position within the brain and

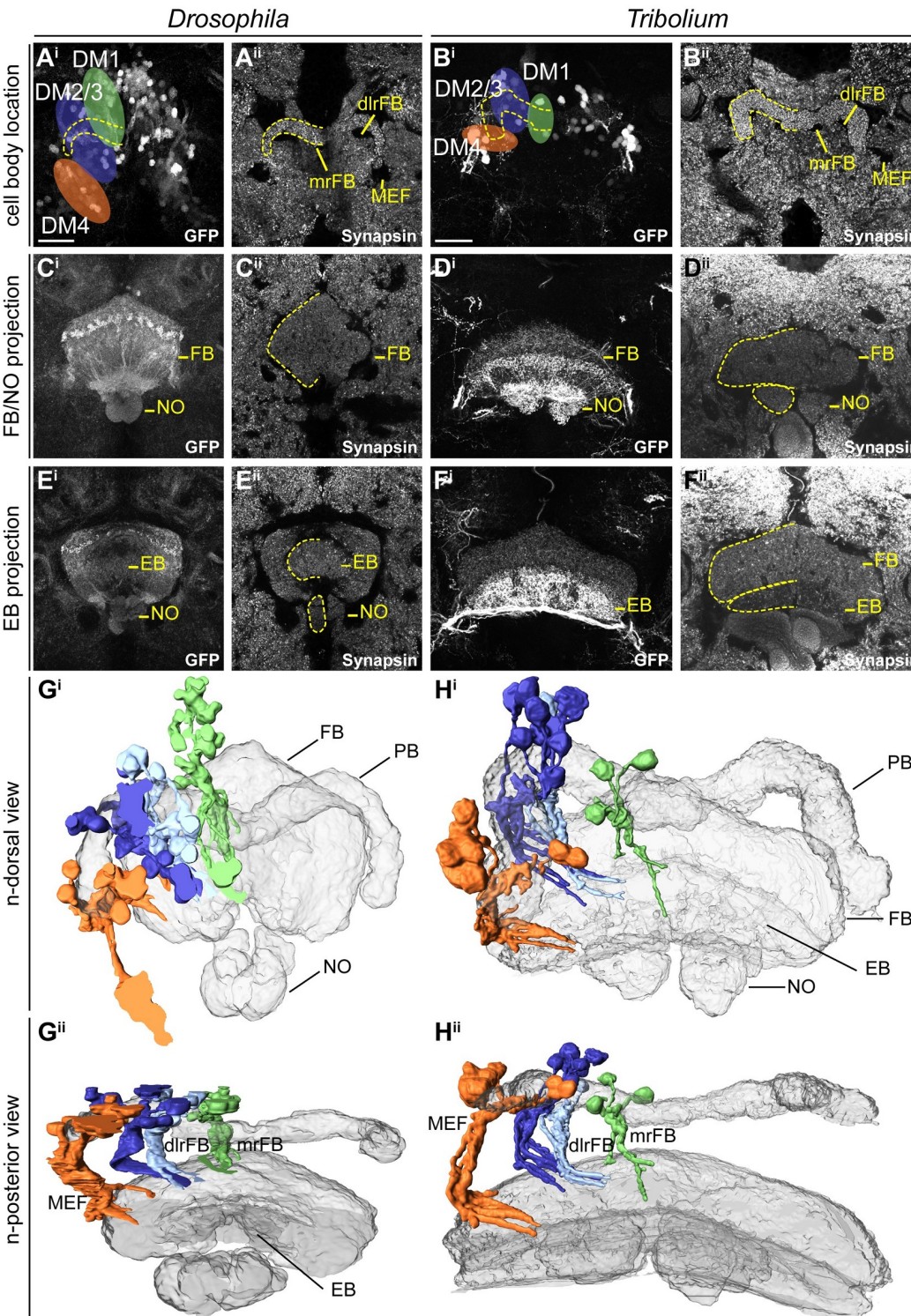

**Fig 3. Homologous Rx-positive cell clusters contribute to the adult central complex columnar neurons of lineages DM1-4.** A to F depict substacks of *Drosophila* (left columns) and *Tribolium* (right columns) adult brains on which the 3D reconstructions in G and H are based. See text for homology criteria. **(A-B)** Cell groups of lineages DM1-4 (colored areas) around the protocerebral bridge (one half marked by yellow dotted line) are shown for *Drosophila* (A) and *Tribolium* (B). **(C-D)** Projection pattern of GFP expressing neurites of these cell groups in the FB and NO. **(E-F)** Much less signal was found in the EB. Note that the *Tribolium* DM4 group had a very high GFP expression level particularly visible in the EB. **(G-H)** 3D

reconstructions of synapsin staining (gray-transparent) and the EGFP marked cells of DM1-4 lineages. $G^i/H^i$ depicts the n-dorsal view shown in A-F. $G^{ii}/H^{ii}$ is rotated to an n-posterior view showing the similarity of the tract architecture. Similar stereotypical positions were found in both species for DM1 (green), DM2/3 (blue shades–sharing a fiber bundle), and DM4 (orange). Because of the large number of labeled cells within the CB, the projections could not be followed further. GFP channels (i) are maximum intensity projections, whereas synapsin channels (ii) are SMEs [66]. dlrFB, dorso-lateral root of the FB; DM, dorso-median; EB, ellipsoid body; EGFP, enhanced green fluorescent protein; FB, fan-shaped body; GFP, green fluorescent protein; MEF, medial equatorial fascicle; mrFB, medial root of the FB; n, neuraxis-referring; NO, noduli; NV, n-ventral; rx, retinal homeobox; SME, smooth manifold extraction. Scale bars represent 25 μm and apply to all panels of each species. https://figshare.com/articles/Fig3/11339798.

with respect to each other. To correctly assign their identities to DM1-4 despite such divergence, we used their projections across the midline as hint and compared the location of the marked cell bodies with recent lineage classifications based on electron microscopy data [12]. Most strikingly, the cell bodies of the DM2/3 lineages were not yet located between DM1 and DM4 (Fig 4E$^i$ and 4K$^i$). In *Tribolium*, in contrast, the DM1-4 cell clusters had an arrangement along the larval PB like the adult situation (Fig 4F$^i$ and 4L$^i$).

The projection patterns of the *rx*-positive DM1-4 lineages differed between the 2 species as well. In *Drosophila*, they formed a straight common projection across the midline in a bundle of parallel fascicles as described before (Fig 4E$^i$, 4G$^i$ and 4I) [12]. Decussation was not found, neither on the level of EGFP signal nor based on acetylated α-tubulin staining (Fig 4G$^i$ and 4I). In *Tribolium*, in contrast, the neurites projected first parallel to the midline toward neuroaxis-posterior (n-posterior, see scheme in Fig 1), projecting through (in the case of DM1-3) or passing by the PB (DM4). Then, they described a sharp turn toward the midline projecting into the lvCB neuropil toward the other side (Fig 4F, 4H and 4L). Basically, this pattern resembled the adult one (compare Fig 4L$^i$ with Fig 3H). In contrast to *Drosophila* L1, acetylated α-tubulin staining (but not EGFP signal) revealed a system of crossing, i.e., decussated fascicles in the region of the lvCB (Fig 4J).

In summary, we confirm that *Tribolium* but not *Drosophila* has a functional CB and PB at the L1 stage. Of note, we interpret the presence of synapsin as an indication for functionality, which, in the case of the *Tribolium* larval CX, is supported by expression of several neuromodulators in specific patterns [9]. However, an unequivocal proof of functionality of particular cells would require EM examination of the brains and electrophysiological recordings. Actually, the *Drosophila* L1 connectome confirms that all DM1-4 cells contributing to the CX indeed lack synapses at the L1 stage [12], i.e., they are most likely not functional.

We further show that the DM1-4 lineages of *Tribolium* larvae already resemble the adult pattern including some decussations, whereas this is not the case in *Drosophila*. Hence, heterochrony is found with respect to the gain of functionality at the L1 stage and with respect to the development of the underlying neural lineages.

Importantly, the functional *Tribolium* lvCB did not yet represent an adult-like upper division. Rather, it morphologically corresponded to a developmental step found in other species as well. Specifically, the initiating decussations within a tract of largely parallel fibers mirrors the situation seen in an embryonic stage of the grasshopper [7,15]. Hence, the *Tribolium* lvCB represents a case of heterochronic gain of functionality of an immature developmental stage rather than a heterochronic shift of the development of an adult-like structure.

## Embryonic CX development proceeds faster in *Drosophila*

Given the heterochronic state found in the L1 larva, we asked how this difference developed during embryogenesis. Specifically, we wanted to test whether the observed differences were due to simple temporal shifts within a conserved developmental series (i.e., classic

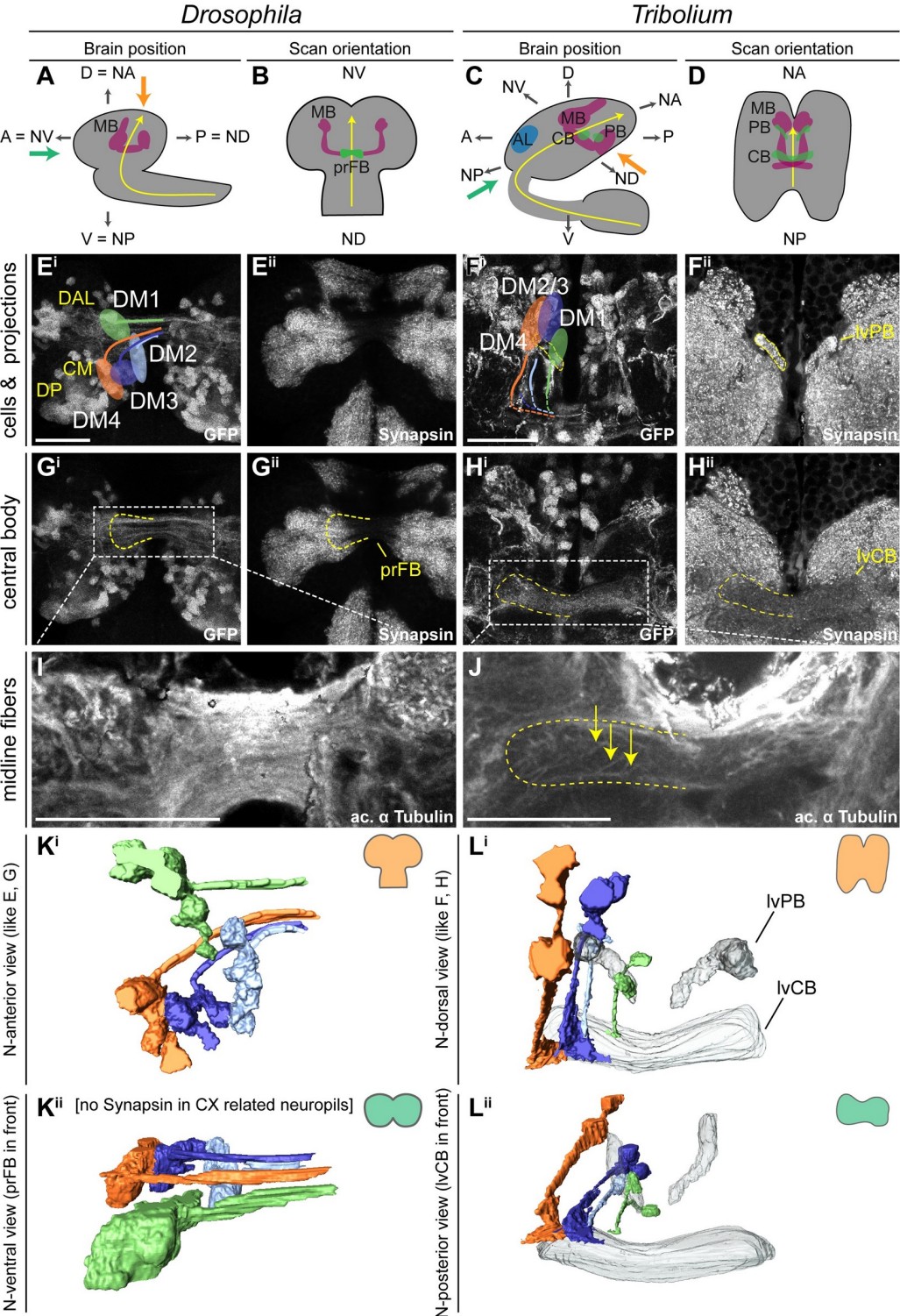

**Fig 4. Different patterns of DM1-4 projection and central complex morphology at the first larval stage. (A+C)** The *Drosophila* (left columns) and *Tribolium* (right columns) L1 brains are positioned differently within the head, visualized by lateral views in A and C. Indicated are the denominators for A, P, D, and V for both body axes and neuraxes (with prefix N). Further shown are the curved neuraxis (yellow) and the larval neuropils MB (magenta), AL (blue), CB, and PB (green). The orange arrows indicate the different directions of the performed scans. The green arrows indicate the orientation displayed in K/L^ii where central complex structures are best visible for both species. **(B+D)** The brains are depicted as they

were scanned in E-J (i.e., from the angle of the orange arrow). **(E-H)** Differences between species were observed in cell cluster position and projection patterns as well as neuropil architecture. In *Tribolium*, arrangement and projection were already similar to the adult (compare L with Fig 3), although the PB was split. In *Drosophila*, it differed dramatically: No CX neuropils were detected, and the DM1-4 lineages projected straight across the midline. In E$^i$, the approximate position of other lineages of the *Drosophila* brain are shown, i.e., DAL, DP, and CM lineages (yellow). **(I-J)** Anti-acetylated-α-Tubulin immunostaining revealed that in *Drosophila* midline-spanning fibers build up a simple stack of parallel fascicles, containing the primordial central body. In *Tribolium*, in contrast, the functional central body contains already some decussated fibers. **(K-L)** 3D reconstructions visualize the spatial relationship between the lineages and highlight the differences between the species. Upper panels (i) reflect the orientation shown in E-H, whereas in the lower panels (ii) are oriented such that the prFB and lvFB are in front, i.e. the central complex is shown in a comparable perspective (see green arrow in A and C). A, anterior; AL, antennal lobe; CB, central body; CM, centro-medial; CX, central complex; D, dorsal; DAL, dorso-anterio-lateral; DP, dorso-posterior; L1, first instar larval; lv, larval; lvFB, larval fan-shaped body; MB, mushroom body (excluding calyx); P, posterior; PB, protocerebral bridge; prFB,; V, ventral. Scale bars represent 25 μm. Original data: https://figshare.com/articles/Fig4/11339804.

heterochronic shift) or whether certain steps of the developmental series had flipped their position within the series (i.e., sequence heterochrony). For this, we compared discrete developmental events of the CX in both *Tribolium* and *Drosophila*. These were the first axon projection emerging from marked cells, the first midline-crossing projection and the stage when a larva-like projection pattern was reached. Further, the emergence of functional CB and PB as judged by synapsin staining was examined. Given the large differences in absolute developmental time between *Tribolium* and *Drosophila*, we used relative developmental time.

The first axons of the *rx genetic neural lineages* formed at a similar relative timing in both species (*Drosophila* 37% developmental time, *Tribolium* 39%; Fig 5A and 5B, see Material and methods, S2 Text and S6 Table for all staging details). The appearance of the first midline-crossing projection appeared earlier in *Drosophila* than in *Tribolium* (*Drosophila* 43%, *Tribolium* 58%; Fig 5C and 5D). Likewise, the "final" larval-like pattern was reached much earlier in *Drosophila* (51%, *Tribolium* 81%; Fig 5E and 5F). At this stage, the tracts of DM1-4 in *Tribolium* showed already an adult-like projection pattern. Moreover, despite an apparently slower pace of development, *Tribolium* performed 2 more steps during embryogenesis, which in *Drosophila* were postembryonic: We found weak decussations and gain of functionality in the prospective CB region (i.e., synapsin staining) in late-stage *Tribolium* embryos at approximately 81% (Fig 5G and 5H). A distinct PB or CB that was clearly differentiated from other areas was not detectable in the embryo, neither in *Tribolium* nor *Drosophila*.

We conclude that both species initiated development of the *rx genetic neural lineage* at a comparable time of development and that *Tribolium* proceeds slower but eventually includes 2 more developmental steps in embryogenesis. This represented a pronounced heterochronic shift of conserved developmental steps between different life stages. More strikingly, certain steps of the developmental series switched their order representing a case of *sequence heterochrony* in brain diversification (Fig 10). Specifically, the decussation and an adult-like tract organization occurred before the larval growth phase of the lvCB in *Tribolium* but after that stage in *Drosophila*.

## In the larva, CX structures grow but do not change basic morphology

Next, we asked how CX structures changed during the larval period from the starting L1 architecture at 50% (Fig 6A–6D) and approximately 95% (Fig 6E–6H) of larval development. In *Drosophila*, the primordium of the FB increased in thickness, particularly after 50% of larval development (compare Fig 6C$^i$ with Fig 6G$^i$), but it remained devoid of synapsin (Fig 6C$^{ii}$ and 6G$^{ii}$) and without decussations. However, the position of DM1-4 cell clusters changed in *Drosophila*. Until 50% of larval development, DM2 and DM3 cell bodies shifted n-ventrally, taking a position between DM1 and DM4 (compare Fig 4E with Fig 6A$^i$). Toward the end of larval

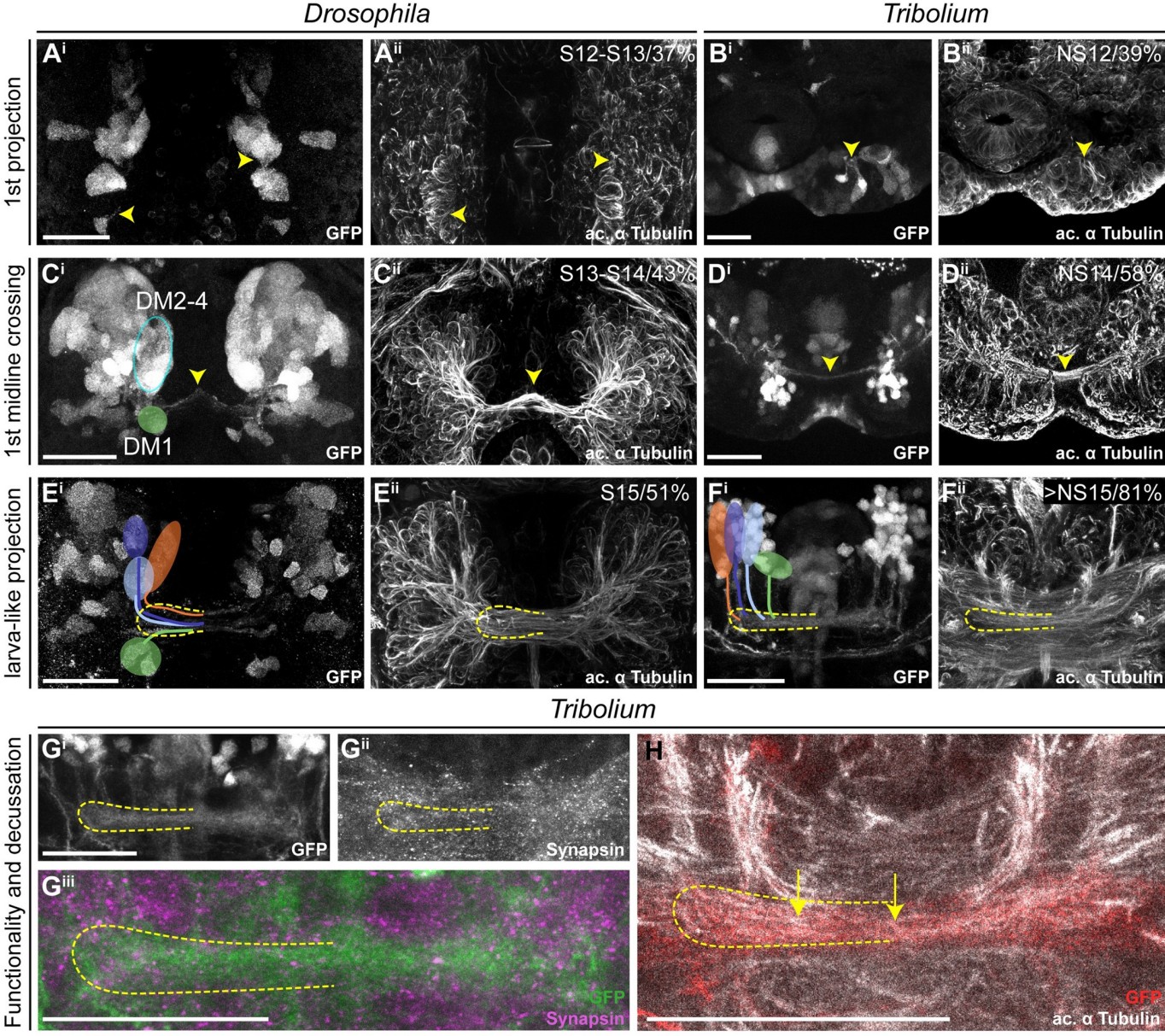

**Fig 5. Key events of central complex development occur during late embryogenesis in *Tribolium* but not in *Drosophila*.** Comparable steps of central complex development are shown in one row for both species, the respective stage/relative time of development are shown in panels ii. The analysis was based on EGFP-labeled neurons (panels i), whereas acetylated α-tubulin staining is shown for reference (panels ii). **(A-B)** The development of the first axons happened at a similar time in *Drosophila* and *Tribolium*. **(C-D)** First midline-crossing fibers appeared earlier in *Drosophila*. **(E-F)** Likewise, the larva-like projection pattern was reached earlier in *Drosophila*. **(G-H)** The late-stage embryonic central complex of *Tribolium* is already faintly synapsin-positive (G^ii, magenta in G^iii), whereas the *Drosophila* lvCB remains synapsin-negative. In *Tribolium*, first decussations were visible (H, yellow arrows). Note that the assignment of *rx*-positive cell clusters to the DM1-4 lineage groups was not unambiguous before midembryogenesis. Tentatively, we indicated the location of DM1 (green) and DM2-4 cells (blue oval form) in C^i. Later, the groups could be assigned to DM1-4 lineages (E-F). Stages in *Drosophila* correspond to [68] and in *Tribolium* to [37]. Posterior is up, except in panels F, G, and H where dorsal is up. Scale bars represent 25 μm. ac, anterior commissure; GFP, green fluorescent protein; lvCB, larval central body; NS, neural stage; Rx, retinal homeobox. Original data: https://figshare.com/articles/Fig5/11339810.

development, cell clusters became arranged in a straight line along the neuraxis, DM1 most n-ventral, DM4 most n-dorsal (Fig 6E^i).

In *Tribolium*, the CB grew in length and thickness as well (compare Fig 6D^i with Fig 6H^i). In addition, the position and shape of the PB changed. In L1 and 50% larval brains, the

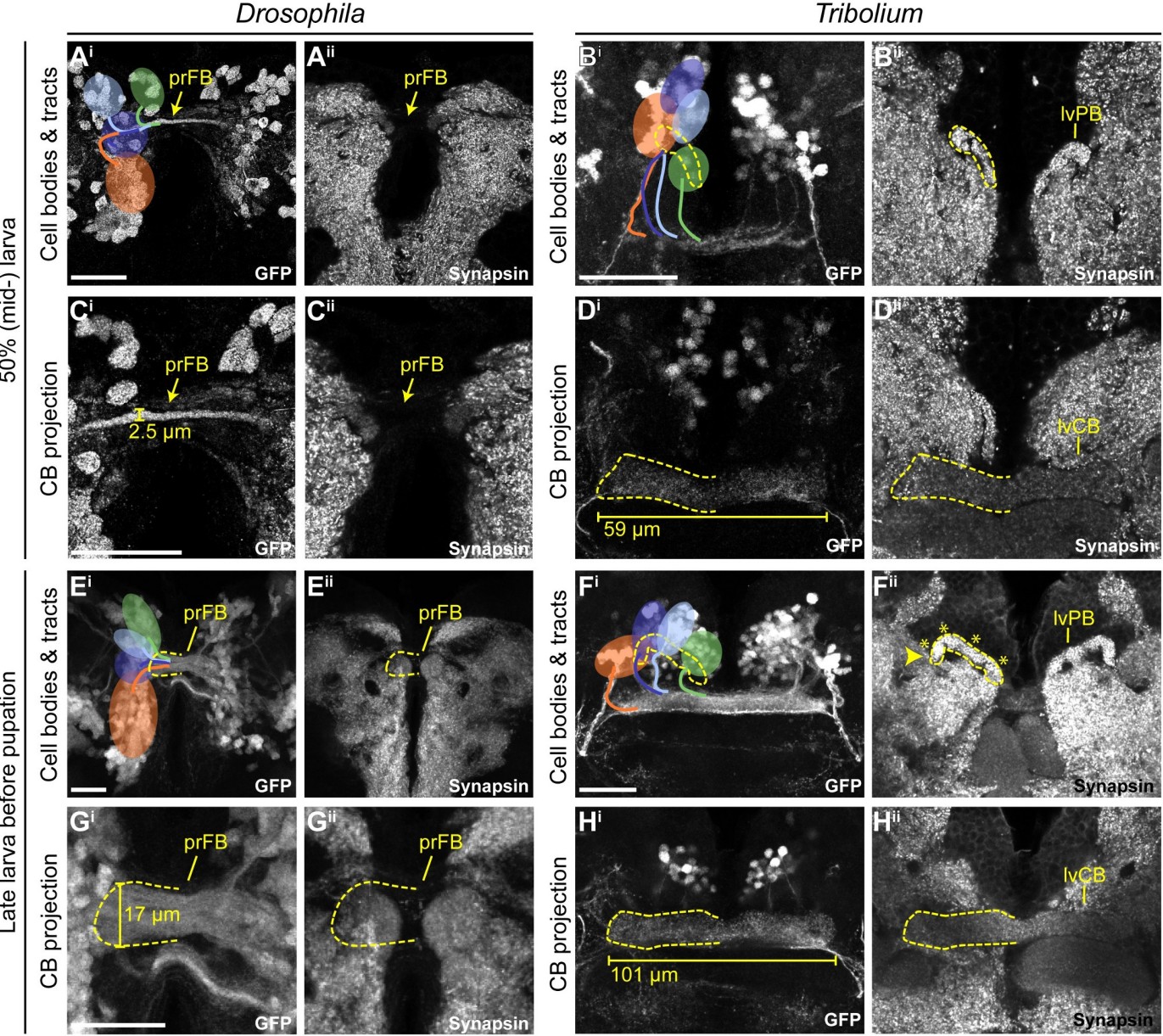

**Fig 6. In both species, the *rx* genetic neural lineage shows substantial growth.** During larval stages, the identified cell clusters and their projections retained their position but proliferated so that larger cell clusters and thicker and larger projections were built. **(A-D)** Depicted are projections at midlarval stages (50% of larval developmental time) in which cell number and projections have qualitatively increased in number and size, respectively. **(E-H)** Shown are late larval stages before pupation, in which cell numbers and projection sizes have increased greatly from 50%. The late lvPB of *Tribolium* can be divided into discrete columns already, indicated by 4 asterisks on one hemisphere. Bars in C, D, G, and H indicate the size increase of midline structures. In *Drosophila*, the prFB increased in width from 2.5 to 17 μm from 50 to 95% of larval development. In L1, the prFB is nondistinguishable using the *rx*-GFP line. The central body of the *Tribolium* L1 brain displayed in Fig 4 was 51.6 μm long, the midlarval lvCB was 58.7 μm, and the late larval lvCB was 100.9 μm long. For *Drosophila* n-ventral and for *Tribolium* n-anterior is up (see Fig 4 for details). Scale bars represent 25 μm and apply to panels i and ii and in case of *Tribolium* to D and H, respectively. GFP, green fluorescent protein; L1, first instar larval; lvCB, larval central body; lvPB, larval protocerebral bridge; n, neuraxis-referring; pr, primordium; *rx*, retinal homeobox. Original data: https://figshare.com/articles/Fig6/11339813.

separate parts of the PB were still oriented along the n-anterior/posterior axis. In late larval brains, however, they shifted into a position more perpendicular to the neuraxis. Accordingly, the positions of the marked cell clusters remained constant in the first half of larval development but then became arranged in one line along the PB from 50% onwards (Fig 6B[i] and 6H[i]). The columnar architecture of the PB appeared to develop during larval stages. Although we did not see columns in L1 (Fig 4F[ii]), we did see them more clearly in late larval stages (Fig 6F[ii]) and probably already in 50% larvae (Fig 6B[ii]). In both species' larval brains, we observed an increase in cell number of the DM1-4 *rx*-positive cell bodies (not quantified).

We conclude that the larval period of CX development is characterized mainly by growth of the CX neuropils in both species. Apart from some shifts of cell body location, the structure established during embryogenesis was mostly maintained during the larval period. Importantly, the *Drosophila* CX precursor remained synapsin-negative, whereas in *Tribolium*, both the lvCB and lvPB remained synapsin-positive, thus still resembling an immature but functional structure throughout the larval period.

## The *Drosophila* CX acquires functionality at later stages of pupal development

Last, we examined pupal stages to reveal when heterochronic divergence in early CX development was eventually leveled out to reach the conserved adult structure. We stained brains of 0% (prepupal stage), 5%, 15%, 20%, 30%, and 50% of pupal development (see S2 Text for staging) for EGFP and synapsin. In *Drosophila*, the PB appeared at 5% of pupal development (Fig 7C[i]), grew subsequently and fused medially between 30% and 50% of pupation (Fig 7I and 7K[i]). Columns became visible at 15% (Fig 7E[i]). The upper division of the *Drosophila* CB appeared first at 5% of pupal development (Fig 7C[ii]). Strength of synapsin staining increased at 15%, coinciding with the emergence of layers and columns structuring the FB (arrows and bars, respectively, Fig 7E[ii]). This coincided with *Dm-rx*-EGFP projections forming a columnar division (Fig 8C[iii]). Thickness increased from 30% onwards resulting in the fan-like structure typical for the *Drosophila* FB (Fig 7G, 7I and 7K[ii]). The *Drosophila* EB emerged later at 15% pupation (Fig 7E[iii]) and continued bending until it formed the typical toroid form that was nearly closed at 50% pupation (Fig 7K[iii]). NO appeared at the same time as the EB as one paired subunit at 15% of pupation (Fig 7E[ii]), and only at 50%, an additional subunit was detected (Fig 7K[ii]). Note that adult NO are eventually composed of 3 to 6 subunits, which apparently developed after 50% development [27].

In *Tribolium*, the larval PB developed further and fused between 5% and 20% (Fig 7D, 7F and 7H[i]; note that we observed a higher heterogeneity in our *Tribolium* dataset with respect to PB fusion and other events). A division into 8 columns typical for the adult PB became visible at 30% on the level of synapses (Fig 7J[i]). However, based on the synapsin and EGFP signal of the *Tc-rx*-EGFP line, a division of the CB into columns was less visible at any developmental stage compared with *Drosophila*. This is in line with previous observations that CX columnar architecture can be visible to quite different degrees in different taxa [1]. Separate upper and lower divisions of the CB became visible already at the beginning of pupation (Fig 7B[ii/iii]). The FB increased in size, and at least 2 layers became visible at 5% (Fig 7D[ii]). The subdivision into columns was faintly visible from 20% onwards (asterisks in Fig 7H[ii]). The EB appeared right at the beginning of pupation with weak synapsin signal intensity (Fig 7B[iii]), which increased from 15% to 20% of pupation (Fig 7F and 7H[iii]). NO appeared at the prepupal stage (Fig 7B[ii]). They thickened considerably at 20% pupation (Fig 7H[ii]), building 2 subunits between 30% and 50% (Fig 7J and 7L[ii]), eventually showing 3 subunits in the adult.

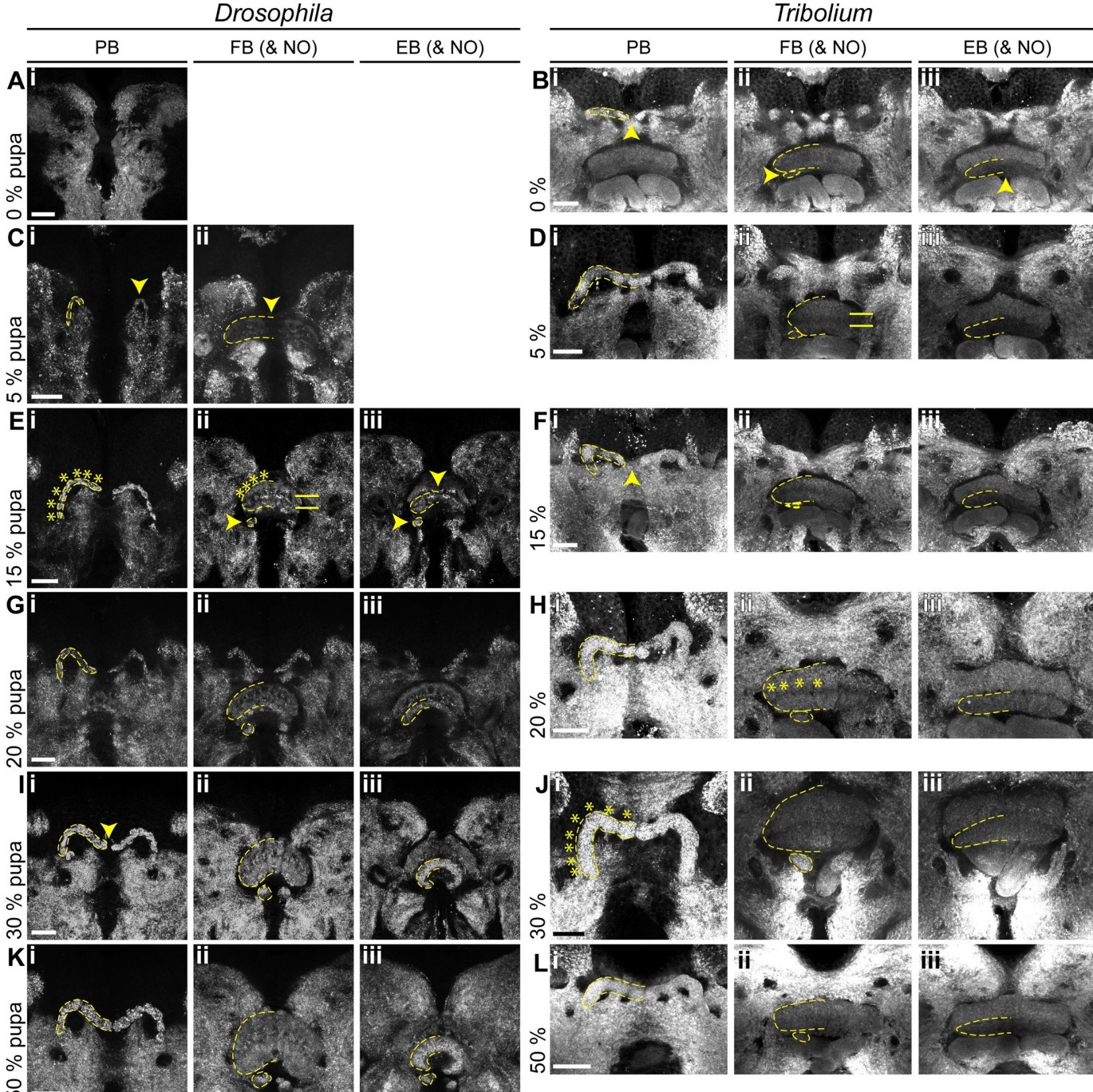

**Fig 7. Pupal central complex development of *Drosophila* is delayed compared with *Tribolium*.** Displayed are substack projections of an anti-synapsin staining of the same preparations used for tracing Rx-positive cell clusters in Figs 8 and 9. **(A-D)** At 0%–5% pupation, in *Drosophila*, the first functional neuropils have appeared, whereas in *Tribolium*, NO and EB have appeared, and the FB developed layers. **(E-H)** At 15%–20% pupation, the *Drosophila* central complex develops columns and layers, and NO and EB appear. In *Tribolium*, columns develop, and the PB fuses. **(I-L)** At 30%–50% pupation, central complex structures resemble the adult, as the PB develops columns and fuses. Note that through slight deviations in positioning of the pupal brains, the FB appears thicker in some stages than in others (e.g., H versus J). The following *Drosophila* events are highlighted by yellow arrowheads: appearance of a functional PB (C^i), FB (C^ii), NO (E^ii/iii), and EB (E^iii), and last stage of an unfused PB (I^i). The following *Tribolium* events are highlighted by yellow arrowheads: the last stage of an unfused PB (B^i, F^i, note the variability in the timing of fusion), appearance of NO (B^ii), and EB (B^iii). A division into distinct layers in the FB are marked by horizontal bars. A division into columns in the PB and FB is marked by asterisks. Scale bars represent 25 μm. EB, ellipsoid body; FB, fan-shaped body; NO, noduli; PB, protocerebral bridge. Original data: https://figshare.com/articles/Fig7-8/11410410 https://figshare.com/articles/Fig7-9/11412777.

We concluded that PB, CB, and NO emerge later in the *Drosophila* pupal brain compared with *Tribolium*. Importantly, during pupation, the *Tribolium* lvCB matures significantly, becoming quite different from its larval appearance.

## The *rx genetic neural lineages* contribute in a similar way to build the CX during metamorphosis in both species

Given the overall heterochronic development of the CX, we asked in how far the development of the *rx genetic neural lineage* reflected these differences during metamorphosis.

In *Drosophila* pupal brains, the array of DM1-4 cell clusters turned from their straight orientation along the midline into a bent configuration following the PB (Fig 8A–8F[i]). The corresponding tracts underwent massive rearrangement, with typical bends similar to an adult configuration already visible at 5% pupation. Most notably, decussations were created by fascicle switching of the DM1-3 tracts starting at 5% of pupal development (Fig 8B[ii]) and became prominent from 15% onwards (Fig 8C[ii]). This resulted in a parallel columnar organisation of the FB at 15% (Fig 8C[iii]) and the marked tracts at 20% (Fig 8D[ii]).

The first EGFP signal clearly corresponding to a FB was found at 15% and 20% (Fig 8C and 8D[iii]) coinciding with the emergence of synapsin staining (Fig 7F[ii] and 7H[ii]). We detected no pronounced projection into the EB until 20%, whereas later projections remained low in intensity (Fig 8F[IV]). Strong projections into the NO were detectable from 15% onwards (Fig 8C[iii]). Following single tracts within the CX was not possible.

In *Tribolium* pupal brains, the cell bodies of the *rx* expressing DM1-4 groups remained comparably similar because they had undergone the respective rearrangement earlier. From 0%–15% onwards, DM1-4 cells formed tracts, which underwent pronounced fascicle switching (Fig 9A–9C[ii]). The resulting division into columns became visible by the presence of strongly marked tracts from 0% onwards in the FB (Fig 9A[iii]) and from 30% in the EB (Fig 9E[iv]).

## Discussion

### Genetic neural lineages as a tool for evolutionary neural development

With this work, we demonstrated that the expression of an orthologous transcription factor can be used as a tool for marking homologous neurons between distant species. Previous efforts have nicely illustrated conservation of certain developmental genes and the cell types and organs they are required for, such as the eye in case of *Pax6/eyeless* [69,70]. Our approach extends the analysis of cell type homology to labeling whole neurons combining regulatory gene expression and projection patterns as arguments for homology of neural cells. Hence, *genetic neural lineages* are very helpful for developmental comparisons. Once homology of a subset of marked cells is confirmed by additional criteria, the timing of developmental stages can be faithfully compared between species (e.g., the first midline crossing). General markers like acetylated tubulin lack this precision because they mark all cells. For instance, the "same" morphologically defined event of a first midline crossing could be performed by nonhomologous cells. However, one should be aware that a *genetic neural lineage* is not equivalent to a *neural lineage* (i.e., one neuroblast and all of its offspring). Rather, we find that several neural lineages are marked both by *rx* (this work) and *foxQ2* expression [41]. In the latter work, we even found indication that both type I and type II neuroblasts may be marked by one *genetic neural lineage*, likely also valid for *rx* (see tentative lineage assignments in SI).

Restricting the marking to fewer cells would be highly welcome. This could be reached by adding a combinatorial component to the system (e.g., marking all cells that express both *foxQ2* and *rx*). Finally, we worked both with an enhancer trap line (*Tribolium*) and a

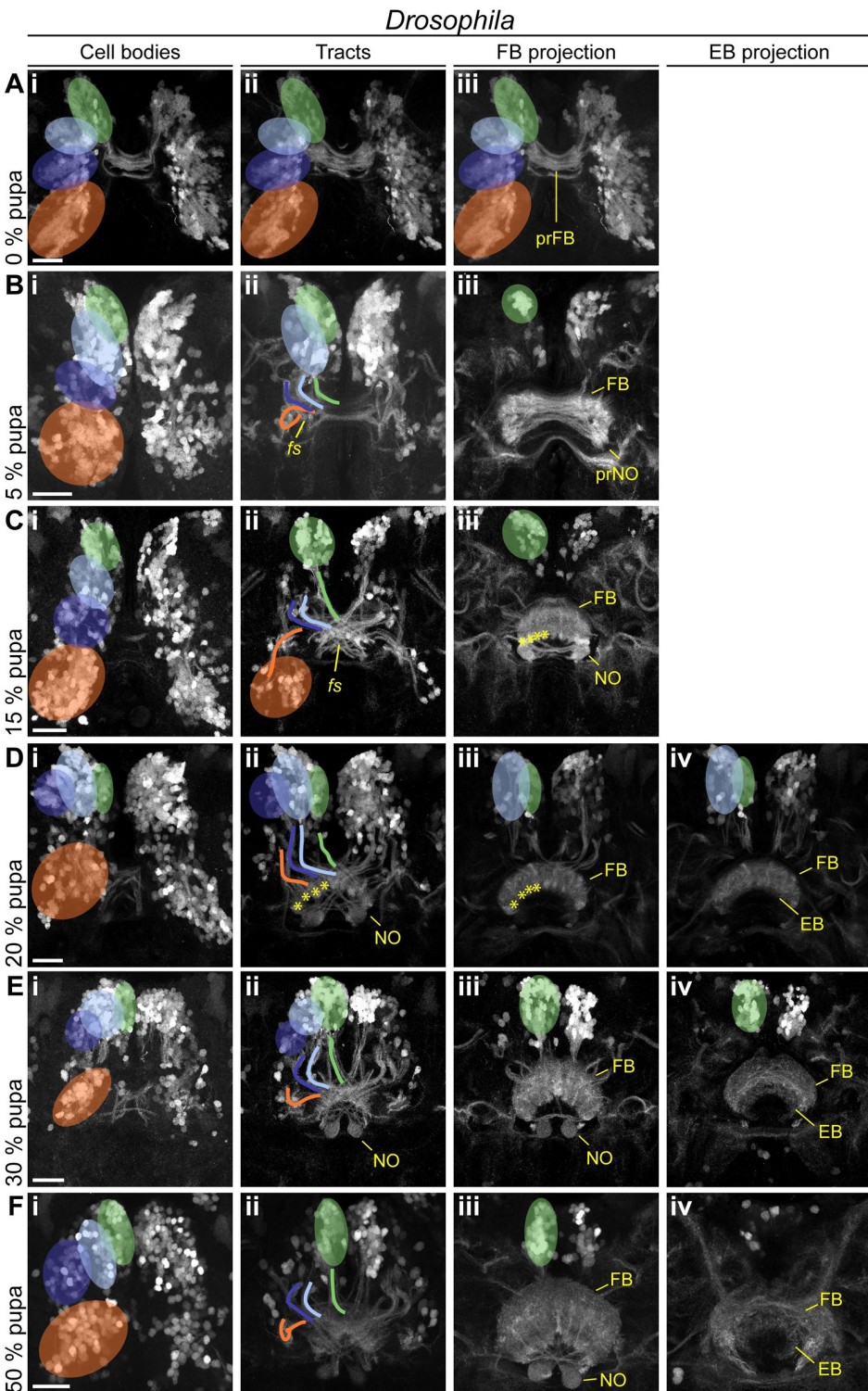

**Fig 8. In *Drosophila*, the main developmental event of fascicle switching with resulting columnar fiber organization occurs in the pupa.** Displayed are subprojections of an anti-GFP staining of the same brain per time point, to display the development and positioning of cell clusters (i) belonging to the DM1-4 lineage and their tracts (ii) (DM1 green, DM2 light blue, DM3 dark blue, DM4 orange) and final projections into the developing central complex neuropils (FB iii, EB iv). **(A-C)** Fascicle switching starts at 5% and is very visible at 15% pupation with the FB and NO developing as result. **(D-F)** Fascicle switching continues, with the EB developing. Also, the cell bodies get

shifted, resembling the shape of the PB as result in later pupal stages. Following events are highlighted: fascicle switching of DM1-3 was visible from 5% onwards (B[ii], C[ii]), with the formation of four columns of the FB per hemisphere (asterisks in C[iii], D[ii], and D[iii]). Scale bars represent 25 µm. EB, ellipsoid body; FB, fan-shaped body; GFP, green fluorescent protein; NO, noduli; PB, protocerbral bridge. Original data: https://figshare.com/articles/Fig7-8/ 11410410.

bicistronic line (*Drosophila*) [39]. The bicistronic line showed high precision in that the overlap with respective antibody staining was almost complete, whereas the enhancer trap showed only a subset, which is in line with known limitations of enhancer trap lines. As long as one restricts the comparison to cells that are homologous by additional criteria (like in this work), this is not critical. However, when a comprehensive comparison of conserved and divergent projection patterns of an entire genetic neural lineage is needed, then bicistronic lines are the better choice. The marking of small but homologous subsets of cells in different species would also be highly welcome for comparative optogenetic experiments.

The fact that our *Tribolium* enhancer trap line did not mark all Tc-Rx-positive cells fortunately does not interfere with our conclusions. First, at developmental stages, 50%–75% of the cells were marked, whereas only in the adult the portion dropped to 10%. Second, we focus on marked cells that are homologous between both species. As we restrict our statements on this marked subset of cells, the presence of nonmarked cells in *Tribolium* does not interfere with our interpretation. Third, only in *Drosophila*, we make a statement on the lack of projection at a certain stage. Luckily, the *Drosophila* bicistronic line showed a 100% overlap of EGFP with Dm-Rx expression, allowing us to make this statement.

## Complex pattern of heterochronies and juvenilization of CX development

Initially, the term *heterochrony* described differences in size and shape emerging mainly from different growth variables such as rate and duration of growth [3,71]. *Sequence heterochrony* was introduced for cases in which certain developmental steps change their position within a developmental sequence [72,73]. To assess the nature and complexity of CX heterochrony, we used 15 events of CX differentiation for which we determined the absolute and relative timing in *Drosophila* and *Tribolium* development (Fig 10).

We find a complex pattern of heterochronies, most of which reflect simple shifts in timing of differentiation events (orange arrows in Fig 10). Interestingly though, some events occur earlier in *Drosophila* (e.g., first embryonic steps 1–3; see Figs 10 and 11 and S5 Table), whereas with respect to others, *Tribolium* develops faster (steps 9 to 13). Importantly, some steps are even shifted between life stages: Formation of adult-like WXYZ tracts and first decussations are embryonic events in *Tribolium* but metamorphic events in *Drosophila* (steps 5–8) (Figs 10 and 11).

We observe that "growth heterochrony" (i.e., different timing, reduction, or prolongation of growth [71]) may not play a major role in CX evolution because most of the growth happens at similar phases in both species (i.e., during early embryogenesis and during the larval stage). This contrasts with the crucial role that growth heterochrony was shown to play in the evolution of brains in other contexts. For instance, in humans, postnatal growth of the brain is strongly increased compared with chimpanzees [74]. Across Mammalia, an increase of proliferation rates probably led to gyrification (folding) of the cortex [75,76]. An intraspecific case of growth heterochrony has been noticed in castes of the honey bee *Apis mellifera*, in which bee queen brains develop faster and as a result, become larger than those of worker bees [77].

Overall, the observed heterochronies reflect a an evolutionary juvenilization along several clades (pedomorphocline) [4]. The *Schistocerca* CX represents the ancestral situation, whereas

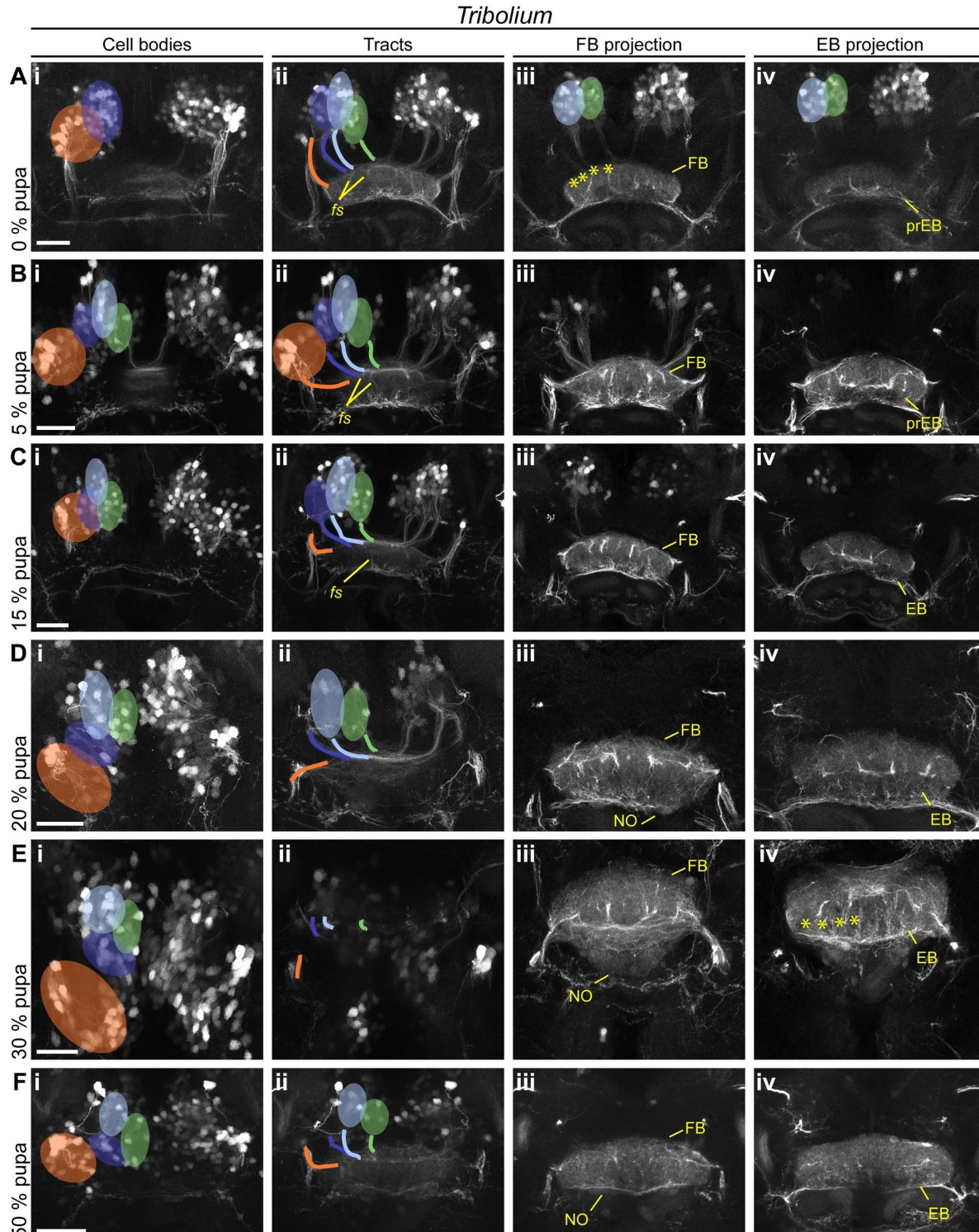

**Fig 9. *Tribolium* pupal development illustrates how the adult central body becomes distinct from the larval form.** Displayed are subprojections of an anti-GFP staining of the same brain per time point, to display the development and positioning of cell clusters (i) belonging to the DM1-4 lineage and their tracts (ii) (DM1 green, DM2 light blue, DM3 dark blue, DM4 orange) and final projections into the developing central complex neuropils (FB iii, EB iv). **(A-C)** Fascicle switching becomes immediately prominently visible at 0% and shows a columnar division of the FB and increases in later stages. **(D-F)** In later pupal stages, decussated projections go into the NO, and a column divided EB. The following events are particularly highlighted: fascicle switching of DM1-3 was visible from 0% onwards (A[ii], B[ii], C[ii]), with a

resulting formation of 4 columns of the FB and EB per hemisphere (earliest visible in A[iii] and E[iv], marked by asterisks). EB, ellipsoid body; FB, fan-shaped body; GFP, green fluorescent protein; NO, noduli; pr, primordium. Scale bars represent 25 μm. https://figshare.com/articles/Fig7-9/11412777.

the *Tribolium* L1 CX is paedomorphic, as it shows similarity to a stage at 60% embryogenesis in *Schistocerca* where decussations have just initiated [33]. Likewise, the *Drosophila* L1 CX is pedomorphic to the *Schistocerca* neuropil, but its primordium equals an even earlier embryonic stage of about 45% to 50% [33], consisting of parallel fibers only.

## An example for sequence heterochrony in brain development

One of our key findings is the presence of sequence heterochrony that contributes to the different forms of larval CX primordia in *Tribolium* versus *Drosophila*. Specifically, adult-like WXYZ tracts, fascicle switching, and gain of functionality of PB and CB (steps 5–8) occur before main net growth of the CB in larvae of *Tribolium*, whereas they occur after this larval growth period in *Drosophila* (Figs 10 and 11). To our knowledge, this is the first example of sequence heterochrony contributing to the evolution of brain diversity. Sequence heterochrony was previously described with respect to processes in which sequences covered for example the entire development of crustaceans [78] or the different order of events of central nervous system, skeletal, and muscular development in Metatheria and Eutheria [79].

The cell behavior underlying sequence heterochrony may be reflected in the development of *pointed*-positive DM1-4 cells in *Drosophila* [12,13]: During embryogenesis, their parallel midline-crossing neurites form the larval FB primordium where they arrest development. Only during late larval and early pupal stages, they continue development building decussations and projections into columns within the FB, forming pontine neurons. Hence, the homologous cells of *Tribolium* would just need to overcome the developmental arrest in order to form first decussations in the embryo. Imaging lines marking the *pointed* genetic neural lineage tailored by genome editing [39] would allow testing this hypothesis.

## Behavioral implications—An immature but functional CX

The CX is essential for orientation and motor control, and the timing of CX development appears to correlate with behavioral demands of the respective life stage. The hatchlings of hemimetabolous species like *Schistocerca* have adult-like legs and compound eyes and need to behave in the same complex environment as the adults. Accordingly, their CX develops fully during embryogenesis. In contrast, the *Drosophila* maggot does not have legs, its visual organs are strongly reduced, and it lives in a rather simple environment, i.e., submerged in food mash. Hence, the absence of a CX in *Drosophila* larvae correlates well with this reduced demand for orientation and locomotion. It was hypothesized that the moving legs of beetle larvae require the presence of a functional upper unit of the CB, whereas the later development of the compound eyes was suggested to require the EB [8,16,17]. The assumption of the presence of a larval FB was based on its bar shape, the presence of tracts presumably prefiguring the lower division and some neuromodulator expression [8,9,11]. Contrasting this view, we find that the functional CB of the L1 does not reflect any adult structure but is a developmental stage in itself that gained functionality (see Results, "Divergent CX structures in the L1 larva of Drosophila and Tribolium" for our definition of functionality). Hence, *Tribolium* has 2 distinct forms of a functional CX, one for the larval and one for the adult life stage.

These findings allow studying CX function from a different angle than before. First, the reduced complexity of the larval CX of *Tribolium* provides a simplified model to study CX

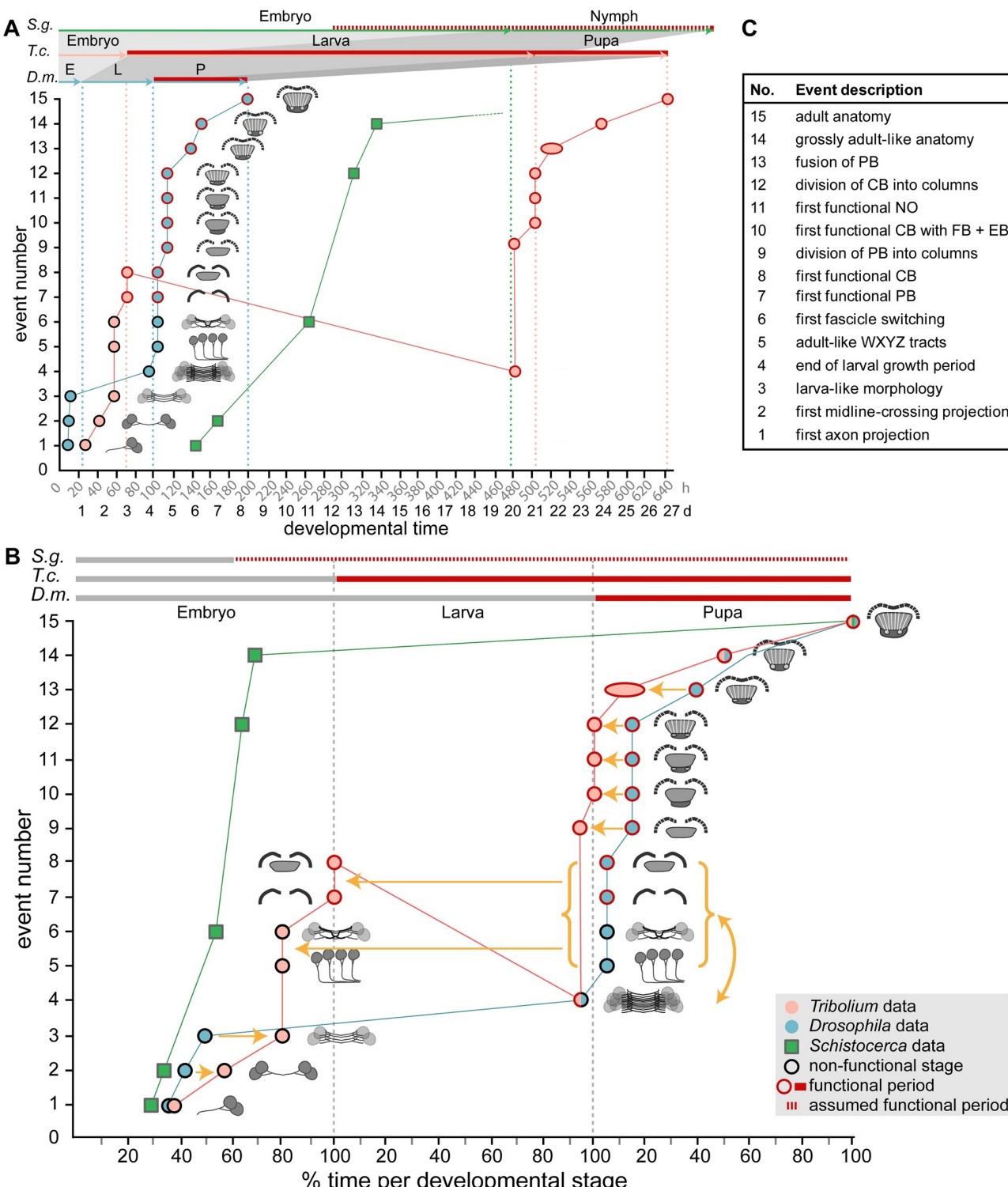

**Fig 10. Schematic summarizing the timing of developmental events of central complex heterochrony.** Developmental time is depicted on the x-axis as absolute time in hours and days (A) or relative time in percentage development of the respective life stages (B). Fifteen discrete events of central complex development (description in C and definition in S5 Table) are depicted on the y-axis and visualized with small sketches. The sequence of events reflects *Drosophila* development. The developmental trajectory shown for *Drosophila* (*D.m.*, blue) and *Tribolium* (*T.c.*, orange) is based on this work, whereas *Schistocerca* (*S.g.*, green) is based on [7,33,80,81]. Red contours of the circles and red lines on the top axes indicate presence of synapsin as a proxy for functionality of the central complex. Synapsin expression data were not available for *Schistocerca*; therefore, neuromodulator expression was used instead (red hatched line). (A) A comparison on an absolute time scale highlights the large differences in actual time between species, and the

resulting divergences over which period a respective animal has a functional central complex neuropil. *Drosophila* has the shortest generation time with the embryonic stage 33%, the larval stage 17% and the pupal stage being 71% of the *Tribolium* time (32˚C in *Tribolium*, 25˚C in *Drosophila*). *Schistocerca* (approximately 31˚C) embryonic central complex development takes more than double of the time of entire *Drosophila* central complex development (480 hours versus 200 hours). **(B)** Initial embryonic development leads to a heterochronic delay in *Tribolium* (orange arrows of events 2 and 3). In *Drosophila*, the larval growth phase follows (4) before in the pupa WXYZ tracts, decussation and gain of synapsin in PB and CB occur (5–8). Strikingly, these latter events are shifted into *Tribolium* embryogenesis. Sequence heterochrony is represented by the developmental sequence 3-**4**-5-6-7-8 in *Drosophila* but 3-5-6-7-8-**4** in *Tribolium* (curved yellow arrow and red line with negative slope). Essentially, the larval growth phase in *Tribolium* occurs after adult-like WXYZ tract morphology and decussation, whereas in *Drosophila*, growth occurs before these events. Pupal events 9 to 13 are heterochronically shifted to earlier stages of development in *Tribolium*. **(C)** Events are shortly described here and defined in S5 Table. CB, central body; EB, ellipsoid body; FB, fan-shaped body; NO, noduli; PB, protocerebral bridge.

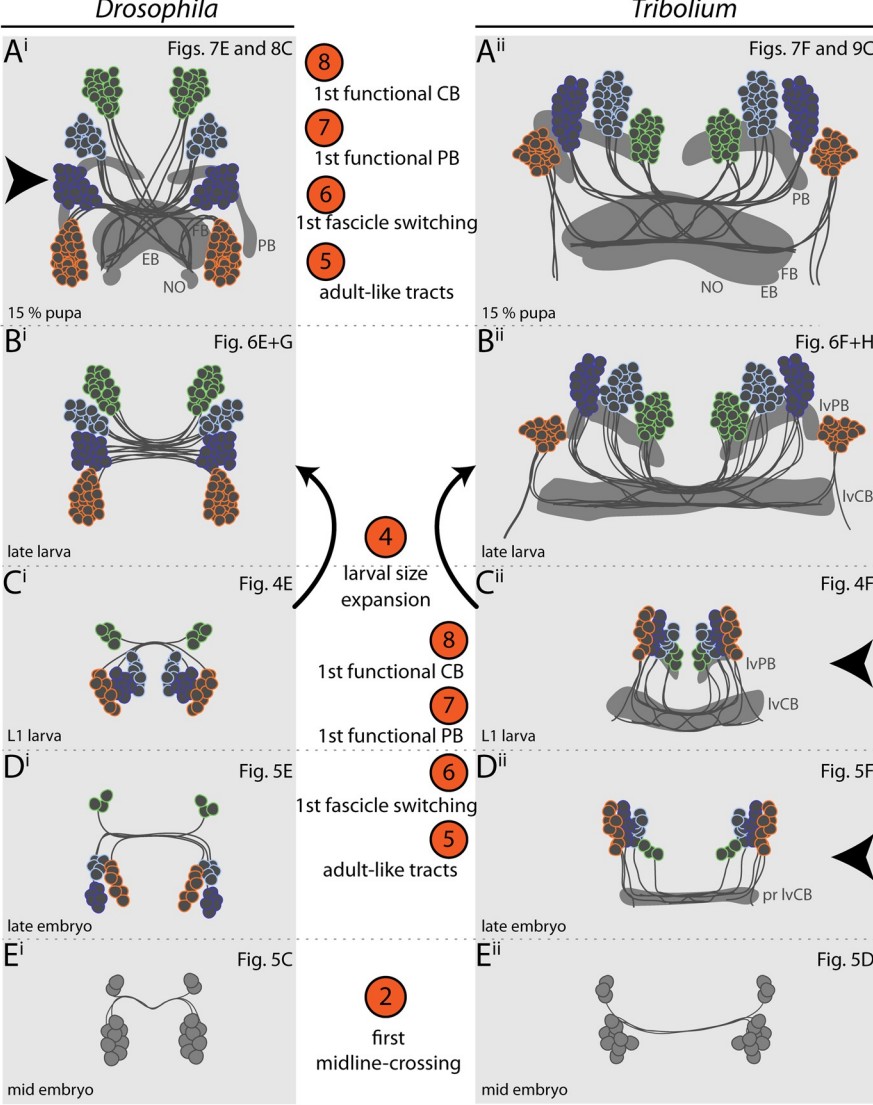

**Fig 11. Sequence heterochrony in central complex development.** All schematics are drawn from the preparations shown in the figures of the results section. Stage numbering corresponds to the one in Fig 10. Steps 2 and 4 (median column) occur at similar developmental stages in both species. However, in *Tribolium* (right panels), steps 5–8 occur before the massive larval expansion of the CX (step 4). In *Drosophila*, in contrast, the larval expansion (step 4) occurs prior to steps 5–8. As consequence, at the end of embryogenesis, the *Tribolium* CX already shows WXYZ tracts, decussations, and synapsin staining. This complex structure grows in size in beetle larvae, whereas in *Drosophila*, a CX primordium of an earlier developmental stage expands. CB, central body; CX, central complex; EB, ellipsoid body; FB, fan-shaped body; NO, noduli.

neural function. The lower number of neurons and the less complex morphology might help understanding the basic neural circuits more easily than in the highly complex adult CX. Specifically, our data suggest that basic functionality appears not to require the separation of upper and lower division of the CB nor a well-developed columnar architecture. Establishing respective genetic and behavioral tools represents a major effort, but the *Tribolium* model system already offers useful technical possibilities like transgenesis, Gal4/UAS system, genome editing, and genome-wide RNAi screening [39,40,45,82,83].

A second open question is the behavioral relevance of the functional larval CX in *Tribolium* larvae. Once tools are established to block the function of the CX exclusively, one could quantify behavioral differences of larvae of both species with intact or disrupted CX, respectively. The prediction would be that the fly larva behavior should remain the same, whereas beetle larvae should reduce their behavioral complexity upon interference. Interestingly, there are genetic data that—while still requiring complete analysis—point into that direction: *Tc-rx* RNAi beetles die at the L1, indicating an essential function of this gene (potentially via the CX), whereas respective *Drosophila* mutants survive until metamorphosis [51].

### The evolutionary scenario—Is a functional larval CX ancestral to Holometabola?

The full development of the CX during embryogenesis in Hemimetabola is ancestral, whereas the evolutionary scenario for holometabolan larvae is more ambiguous. We argue that the presence of a functional larval CX (like in *Tribolium*) is ancestral, whereas its loss (like *Drosophila*) is derived. At first sight, our comparison between these 3 species would indicate that flies have retained the ancestral condition, whereas beetle and other holometabolous insect larvae have gained functionality as evolutionary novelty. This is based on the observation that the developmental steps occur in the same order in *Drosophila* and *Schistocerca*, whereas the sequence heterochrony is observed in *Tribolium* (Fig 10).

However, this scenario requires assuming independent gains of functionality of larval CXs in several holometabolous insect taxa, as these are not only observed in beetles [8]. Moreover, the larvae of beetles, which have legs and move and orient in an environment using eyes, are thought to be more representative of the ancestral holometabolous insect larva than the *Drosophila* maggot with its reduction of legs, eyes, and head [84]. Therefore, we prefer the alternative scenario, which puts the emergence of a functional larval CX at the basis of the evolution of Holometabola. Indeed, besides the CX, larvae have a number of immature but functional organs. For instance, they have simplified legs, antennae, and eyes, whereas other organs lack completely (e.g., wings) [84]. Given the necessity of the larva to orient itself using simple eyes and to behave using their legs, a minimal functionality of the CX might have been a prerequisite for the evolution of the larval stage. In this scenario, the occurrence of larval functional CXs in several taxa would reflect conservation, whereas the lack in *Drosophila* and other larva would reflect a loss as evolutionary divergence. Indeed, the fly maggot may need less elaborate orientation behavior because it hatches within the food source that usually supports its entire development.

## Material and methods

### General considerations

We adhered to the nomenclature presented in [34], except for our reference to the DM4 ipsilateral fascicle as tract, which we remain with the term W tract [15]. We refer to the divisions of the CB as FBs and EBs for sake of comparability with *Drosophila* literature. Where

morphologically meaningful, we use the synonyms upper and lower division, which are terms used in classic insect literature. We use the traditional term "columns" for vertical subdivisions in the CX, whereas "slices" has been suggested as synonym [34].

Animals were kept at 32°C for *T. castaneum* and 25°C for *D. melanogaster* under respective standard conditions [85,86]. Except for embryos and young larvae where sexing was not possible, females were selected for stainings. Besides in Fig 5G ($N = 1$), Fig 5H ($N = 2$), and Fig 6B–6D ($N = 2$), the dataset consisted of at least $N = 3$ tissues. All stacks from which figures were created, and films in .avi format thereof can be found under Figshare (https://figshare.com/projects/Additional_data_for_Farnworth_et_al_/64799). All *Drosophila* and *Tribolium* stocks, antibodies, and dyes as well as primers are documented in S2–S4 Tables. Detailed information on all methods used can be found in S2 Text.

## Tc-Rx antibody generation and verification

The anti-*Drosophila* Rx antibody was kindly gifted by Dr. Uwe Walldorf [51]. No cross reactivity to the Tc-Rx protein was found. Hence, we generated an antibody against Tc-Rx by cloning the region N-terminal to the homeobox domain into a GoldenGate vector containing a SUMO peptide (KNE001, S1 Vector, S2 Text), expressing it in BL21-DE3 Rosetta bacteria and purifying it by immobilized metal ion affinity chromatography. A guinea pig antibody was then raised against the purified peptide by Eurogentec (Kaneka Eurogentec S.A., Belgium). Finally, specificity of the antibody was verified by in situ hybridization against *rx* RNA combined with Tc-Rx immunostaining as well as immunostaining of *Tc-rx* RNAi-mediated knockdown embryos (S1 Fig).

## *rx*-EGFP transgenic lines

For *Drosophila*, a trangenic line marking large parts of *rx* expression was not available. Therefore, we generated a bicistronic line by CRISPR/Cas9-mediated homology-directed repair (S3 Fig) [39]. Toward this end, we removed the endogenous STOP codon of the *rx* ORF to generate an in-frame *rx-EGFP* fusion gene. In addition, we included a sequence encoding for the P2A peptide between the *rx* ORF and *EGFP* CDS to ensure that 2 distinct proteins from a common RNA will have been translated (for information on the P2A peptide, see [61,62]). We also included an eye marker allowing us to screen $G_1$ positives with ease. The repair template was cloned using the Gibson assembly kit (New England Biolabs, MA, USA). Suitable target sites without off-targets were identified using the CRISPR Optimal Target Finder [87] (http://targetfinder.flycrispr.neuro.brown.edu/). Respective guides were cloned into an U6:3-*Bbs*I vector and subsequently tested by a T7 Endonuclease I assay. The repair template and guideRNA containing plasmids were co-injected into *Act5C-Cas9*, *DNAlig4[169]* embryos [88]. Surviving $G_0$ animals were crossed individually to $w^-$ virgins of the opposite sex, and the $G_1$ generation was screened for eye marker and EGFP reporter. The overlap of EGFP and Rx was determined by double immunostainings in adults and embryos. Indeed, we found that each cell expressing Rx now also expressed EGFP, largely located in the cytoplasm.

For *Tribolium*, we identified a suitable transgenic line in the GEKU base website where its insertion had been mapped to the upstream region of *Tc-rx* (# E01101, http://www.geku-base.uni-goettingen.de/, S2 Fig) [47]. This *Tc-rx*-EGFP line was verified by Rx/GFP co-immunostainings, which revealed that all EGFP-expressing cells also expressed Rx (with the exception of the eye transformation marker). As with most enhancer traps, the resultant pattern did not exactly mirror the expression of *rx*. Quantification revealed the overlap of Tc-Rx and EGFP signal in DM1-4 cells to be 9.71% (STD ± 1.81) at the adult stage, 77.39% (STD ± 9.98) at the L1 stage, and 50.23% (STD ± 4.39) in the prepupal brain (see S2 Text, S2 Fig).

Both *Dm-rx*-EGFP and *Tc-rx*-EGFP were made homozygous, and all data used derives from homozygous stocks.

### Comparative staging and determining CX events

A description of the stages that we defined are documented in S2 Text and S5 Table. Exact values for the timing of CX developmental events displayed in Fig 10 are found in S5 Table.

### Fixation, staining, imaging, and image processing

Fixation, in situ hybridization, and immunostainings were performed as described in [38,42] with details in the S2 Text and S7 Table. Images were taken with a Leica SP8 confocal microscope (Wetzlar, Germany) with standard settings. Images were examined using Fiji software (open source software; https://fiji.sc/) [89]. The 3D reconstructions were performed using Amira 5.4.1 (Visage Imaging, Fürth, Germany), and figures were created using Adobe Illustrator CS5 (Adobe Systems, San José, CA, USA).

## Supporting information

**S1 Text. Supporting results.**
(PDF)

**S2 Text. Supporting material and methods.**
(PDF)

**S1 Vector. KNE001 (pET SUMO-GoldenGate).** Vector used for protein expression.
(GB)

**S2 Vector. MF01 (pJet1.2-Dm-rx_bicistronic-construct).** Repair template for the Dm-Rx-GFP bicistronic construct. In the construct, we included an insect codon-optimized version of the P2A peptide, with the following sequence: GGGTCCGGCGCCAC-CAACTTCTCCCTGCTGAAGCAGGCCGGCGACGTGGAGGAGAACCCCGGCCCC.
(GB)

**S1 Fig. Generation and validation of the Tc-Rx antibody. (A)** Alignment (Geneious 11.1.5, Geneious Alignment) of Rx proteins of *Drosophila* and *Tribolium* as well as representative species. The conserved homeobox and OAR (O) domains (gray) are present in all proteins. Antigenic regions for the Dm-Rx [51,53] and the Tc-Rx antibody are displayed in magenta. The Dm-Rx protein was shortened for better display (amino acids 1 to 200 and most between 800 and 900 are not displayed). Notice that the *D. melanogaster* antigenic region appears to be absent in *T. castaneum* and all other species. **(B-C)** Tc-Rx protein and *Tc-rx* RNA expression in *Tribolium* embryos of neurogenesis stages 3 and 11 [37] were depicted (Zeiss LSM510, 40× immersion objective) as maximum intensity projections (DAPI for structure as average projection). Anterior is to the top. Animals were mounted dorsal up. The signal detected in the antibody staining against Tc-Rx protein (magenta) overlapped to a high degree with the signal detected in the in situ hybridization (green). Note that although the protein of Tc-Rx was located in the nucleus, *Tc-rx* RNA was also in the cytoplasm of the cell soma, which resulted in a different cellular localization. **(D)** To validate the specificity of the Tc-Rx antibody, we performed a RNAi-mediated *Tc-rx* knockdown. Indeed, Tc-Rx expression was reduced in knockdown embryos. Depicted are 3 categories of Tc-Rx expression (i.e., Tc-Rx antibody staining intensity, magenta, as maximum intensity projections) after knockdown (strong, equaling wildtype, in D[i], intermediate in D[ii], weak in D[iii]). To accommodate for differences in intensity of staining, a co-staining against Invected/Engrailed with the respective antibody was

performed. **(E)** A total of 34 RNAi embryos were categorized into the 3 expression intensity groups in a blinded experiment. Wild-type animals showed a high level of expression and were mostly grouped in category "strong" with some in category "intermediate." No knockdown animals were grouped into the "strong" category, most in "intermediate" and some in "weak" (Fisher's exact test, $P < 0.001$). Scale bars represent 100 μm.
(TIF)

**S2 Fig. Characterization and validation of *Tribolium rx-EGFP* enhancer trap line. (A)** The *Tribolium rx-EGFP* enhancer trap was taken from the GEKU screen collection [47] where enhancer traps were generated by *piggyBac*-mediated transposition. A *3XP3-EGFP-SV40* cassette randomly inserted upstream of the *Tc-rx* gene in opposite direction (insertion site mapped by [47]). **(B)** Maximum intensity projections of immunostainings against GFP and Tc-Rx in adult brains of the *Tc-rx*-EGFP line. The line only marked a subset of Tc-Rx expressing cells. This also applies to the n-dorsal region (B[ii]). However, all EGFP-expressing cells also expressed Tc-Rx. Coexpression was verified manually. **(C)** The introduction of the enhancer trap cassette did not visually influence Tc-Rx expression, as domains were highly similar between transgenic Rx-GFP (B[i]) and wild-type *vermillion-white* (*v[w]*, B[ii], [90]) animals, as visualized by color-coded maximum intensity projections. Observed qualitative differences in Tc-Rx expression in the transgenic or wildtype condition (*N* = 3 each) were approximately as large as the differences between the genetic backgrounds. **(D)** A crop of a maximum intensity projection of cells surrounding the adult protocerebral bridge (yellow arrowhead, D[i]) shows the coexpression of GFP (D[ii]) and Tc-Rx (D[iii]) in a subset of cells that were subsequently used in this study. **(E)** An analogous analysis in young pupal brains of cells surrounding the protocerebral bridge (E[i]) revealed more EGFP-expressing cells (E[ii]) with overlap to Tc-Rx cells (E[iii]) than in the adult (D). **(F)** Quantification of Rx/GFP double-positive cells in the region of DM1-4 lineages surrounding the protocerebral bridge (yellow rectangle, F[i]) revealed that at different developmental stages, the fraction of double-positive cells is different, ranging from approximately 10% to 75%. Scale bars in B and C represent 100 μm, and in D and E, scale bars represent 50 μm. EGFP, enhanced green fluorescent protein; GFP, green fluorescent protein; Rx, retinal homeobox protein.
(TIF)

**S3 Fig. Strategy, generation, and validation of *Drosophila* bicistronic *rx*-EGFP transgenic line. (A[i])** Strategy of building a *Dm-rx*-EGFP line (modified from [39]). Two gRNAs next to the endogenous STOP codon (guide A, brown dashed line) and downstream of the *Dm-rx* 3'UTR (guide B, blue dashed line) were used. The DNA repair template included a sequence encoding for a *P2A* self-cleaving peptide, *EGFP*, the endogenous region between guide A and B (*Dm-rx* 3'UTR and a fraction of intergenic region), and the *3xP3-DsRed-SV40* eye marker, as well as 1-kb homology arms flanking the insertion sites. **(A[ii])** The edited transgenic locus comprises a common open reading frame of both *Dm-rx* and *EFGP* with a STOP after *EGFP*. **(A[iii])** Four gRNAs were used in different combinations to generate similar transgenic lines. The gRNAs used for the transgenic line used in this study are marked in bold (guide A and B3). **(B)** Overview of gRNA sequences and transgenesis statistics upon injection [88] for the *Drosophila* Rx-GFP transgenic line. **(C[i])** Immunostaining of anti-Dm-Rx (magenta) and anti-GFP (green) in the *Dm-rx*-EGFP line showed that all visible cells that expressed Dm-Rx also expressed GFP, shown in a smooth manifold extension (SME) projection [66] of a brain hemisphere of a S16 embryo. The region marked with a dotted line in C[i] is shown in **(C[ii])** as a single slice. Here, the different cellular localizations are visible. Dm-Rx retained its nuclear localization, while GFP located to the cytoplasm, demonstrating functionality of the P2A peptide. **(D)** The transgenic line had normal Dm-Rx expression, shown by anti-Dm-Rx immunostaining and depth color-coded maximum intensity projection in the Rx-GFP line (D[i]) and the origin

wildtype strain w[1118] (D[ii]). Observed qualitative differences in Dm-Rx expression in the transgenic or wildtype condition (*N* = 3 each) were approximately as large as the differences between the genetic backgrounds. **(E-F)** Dm-Rx and EGFP expression matched in adult brains (see yellow arrowheads for exemplary double-positive areas). Maximum intensity projections of synapsin immunostainings (E[i], F[i]), GFP (E[ii], F[ii]) and Dm-Rx (E[iii], F[iii]) in an adult *Drosophila* brain. Anti-synapsin (E[i], F[i]) marked brain position. E is n (neuraxis)-ventral and F is n-dorsal [34]. Scale bars in D-F represent 100 μm and in C 25 μm. EGFP, enhanced green fluorescent protein; GFP, green fluorescent protein; gRNA, guide RNA; Rx, retinal homeobox protein; SME, smooth manifold extraction.
(TIF)

**S4 Fig. Conserved expression of Rx protein in the adult brain of *D. melanogaster* (A, C) and *T. castaneum* (B, D) as well as lineages marked by Rx expression.** We mapped the labeled Rx-positive cells to previously described lineages of the *Drosophila* brain using locations relative to other brain structures and their projection pattern as criterion ([63, 64]; www. mcdb.ucla.edu/Research/Hartenstein/dbla/index.html and references therein). We tentatively named *Tribolium* cell clusters by using similar locations and projections as compared with the *Drosophila* atlas, used as guide. A list of all lineages with names and descriptions can be found in S1 Table. Hemispheres are separated by a red dotted line for orientation. Because of the cell body rind expression of Rx, domains and proposed lineages could be separated into 2 fractions, n-ventral and n-dorsal, corresponding to each half of the insects' brains. For each species, 1 image stack was used and separated into 2 fractions. Rx expression is displayed by a maximum intensity projection of a substack of an anti-Rx immunostaining (i). Basic anatomical structure of the insects' brains is displayed by a SME projection [66] of a synapsin immunostaining (ii). On this projection, in the left hemisphere, the locations of the proposed lineages are shown color-coded, whereas on the right hemispheres, basic anatomical structures are annotated that assist understanding differences in domain position between the species (yellow). dlrFB, dorso-lateral root of the FB; LAL, lateral accessory lobes; MEF, medial equatorial fascicle; ML, medial lobe; mrFB, medial root of the fan-shaped body; PB, protocerebral bridge; PED, peduncle; VL, vertical lobe. Scale bars represent 100 μm.
(TIF)

**S5 Fig. Previously described *pointed*-positive cells of the central complex are a subset of Dm-Rx-positive cells.** Displayed is a co-localization of Dm-Rx-positive neural cells and cells under the control of R45F08-GAL4 [13,91] shown in brains of *Drosophila* wandering third instar larvae. **(A-B)** Antibody staining in a cross of the R45F08-GAL4 line and UAS-mCD8:: GFP was performed against Dm-Rx (depicted in magenta) and GFP (green) to reveal the coexpression of cell bodies of lineages DM1-3/6, marked through the R45F08-GAL4 line, and Dm-Rx. Approximately 90% of the R45F08-GAL4 GFP-positive cells were Dm-Rx-positive as well (A-A[ii] first half, B-B[ii] second half of the stack). **(C)** Antibody staining in animals (*N* = 2) of the respective cross from subcrosses of the Rx-GFP line each with R45F08-GAL4 line and the UAS-mCD8::RFP (SMEs, see [66]). This resulted in a coexpression of GFP in a Dm-Rx expression pattern and RFP under control of R45F08-GAL4. Antibody staining against GFP (cyan) and RFP (red) revealed coexpression of both fluorescent proteins in midline crossing projections. Although RFP is membrane-bound and GFP cytoplasmic, there were several fascicles showing coexpression of RFP and GFP. This corroborated the high degree of overlap of Dm-Rx and DM1-3/6 lineage offspring shown in panels A and B. Scale bars represent 50 μm. GFP, green fluorescent protein; Rx, retinal homeobox protein; SME, smooth manifold extraction.
(TIF)

**S1 Table. Proposed lineages expressing Rx in the adult *Drosophila* (*Dm*) and *Tribolium* (*Tc*) brain.** Listed are 11 lineages with identifier, name, and a description relative to the neuroaxis, as well as the position in Fig 2 and S4 Fig and the degree how unequivocally the assignment of their stereotypical projections was. Identification of lineages is based on [63, 64], https://www.mcdb.ucla.edu/Research/Hartenstein/dbla/index.html, and references therein. AVLP, anterior ventrolateral protocerebrum; CA, calyx; LAL, lateral accessory lobes; MEF, medial equatorial fascicle; PB, protocerebral bridge; PED, peduncle; SLP, superior lateral protocerebrum; SMP, superior medial protocerebrum.
(XLSX)

**S2 Table. Primer list.** P1 to P12: black writing—annealing part, red—overlapping part, green —PAM modification.
(XLSX)

**S3 Table. List of primary and secondary antibodies as well as dyes used in this study.**
(XLSX)

**S4 Table. *Drosophila* and *Tribolium* stocks used in this study.**
(XLSX)

**S5 Table. Description and definition of 15 central complex related events used in this study to illustrate heterochronic development in *Tribolium castaneum* (*Tc*), *Drosophila melanogaster* (*Dm*) and *Schistocerca gregaria* (*Sg*).** Events were defined by using our dataset of anti-GFP and anti-synapsin stainings with both species, to determine potential differences between them, and by using the central complex literature as reference point. Time points for each event are included, as absolute time in hours and relative time per developmental period in percent. Note that length of embryonic developmental periods was taken from [37,68] and Scholten and Klingler (unpublished), stages were determined using morphological criteria and then time points were calculated from these works. Larval and pupal developmental times were determined specifically for our *rx* transgenic lines and pupal time points were then verified by morphological criteria using works by Dippel (unpublished) and [92]. Information on *Schistocerca* central complex events as well as relative and absolute developmental time was taken from [7,33,80,81]. CB, central body, *Dm*, *Drosophila melanogaster*; EB, ellipsoid body; FB, fan-shaped body; NO noduli; PB, protocerebral bridge; *Sg*, *Schistocerca gregaria*; *Tc*, *Tribolium castaneum*.
(XLSX)

**S6 Table. Stages and their definition included in this study.**
(XLSX)

**S7 Table. Immunohistochemistry in stages (excluding embryos) of both species.** There are 2 variations of adult stainings. Antibodies were used as in S3 Table except for synapsin. NGS, normal goat serum; PB, phosphate buffer [93]; PFA, paraformaldehyde; T, Triton-X-100 with % in PB.
(XLSX)

## Acknowledgments

Dr. Felix Quade helped with 3D reconstructions, and Lara Markus provided some embryonic and larval immunostainings. We thank Prof. Uwe Walldorf for providing the Dm-Rx antibody and Prof. Christian Wegener for providing the anti-Synapsin antibody. Dr. Achim Dickmanns supported protein expression and purification. Analyses of brain anatomy and homologous

cell group identification were supported by Prof. Volker Hartenstein. We want to further thank Dr. Stephen H. Montgomery and Prof. Robert A. Barton for fruitful discussions. We thank Drs. Marita Büscher and Nico Posnien for valuable discussions as well as Stefan Dippel for sharing unpublished data on pupal staging, Elke Küster and Claudia Hinners for technical support, and Dr. Elisa Buchberger for helpful corrections of the manuscript.

## Author Contributions

**Conceptualization:** Max S. Farnworth, Gregor Bucher.

**Formal analysis:** Max S. Farnworth.

**Funding acquisition:** Gregor Bucher.

**Investigation:** Max S. Farnworth.

**Methodology:** Max S. Farnworth, Kolja N. Eckermann.

**Project administration:** Gregor Bucher.

**Supervision:** Gregor Bucher.

**Visualization:** Max S. Farnworth, Gregor Bucher.

**Writing – original draft:** Max S. Farnworth, Gregor Bucher.

**Writing – review & editing:** Max S. Farnworth, Kolja N. Eckermann, Gregor Bucher.

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
