## [Editor Report · Decision Letter 0]

6 Jan 2020

Dear Dr Bucher, 

Thank you for submitting your manuscript entitled "Sequence heterochrony led to a gain of functionality in an immature stage of the central complex: a fly-beetle insight" for consideration as a Research Article by PLOS Biology.

Your manuscript has now been evaluated by the PLOS Biology editorial staff as well as by an academic editor with relevant expertise and I am writing to let you know that we would like to send your submission out for external peer review.

Please re-submit your manuscript within two working days, i.e. by Jan 08 2020 11:59PM.

Kind regards,

Ines

--

Ines Alvarez-Garcia, PhD

Senior Editor

PLOS Biology

Carlyle House, Carlyle Road

Cambridge, CB4 3DN

+44 1223–442810

---

## [Decision Letter · Decision Letter 1]

8 Apr 2020

Dear Gregor,

Many thanks for your patience while we reconsidered our decision to decline your manuscript entitled "Sequence heterochrony led to a gain of functionality in an immature stage of the central complex: a fly-beetle insight" for publication as a Research Article at PLOS Biology.

I have now discussed your appeal and rebuttal with the team editors and the Academic Editor, and we do feel that your responses are promising and potentially convincing at addressing the previous points raised by the reviewers. Therefore, we have decided to reset the decision and allow you to submit a revised version of the manuscript along the lines you pointed out in the rebuttal, addressing all the specific points and in particular the heterochrony.

Please be aware that we cannot make any decision about publication until we have seen the revised manuscript and your response to the reviewers' comments. Your revised manuscript is also likely to be sent for further evaluation by the reviewers.

We expect to receive your revised manuscript within 2 months. 

**IMPORTANT - SUBMITTING YOUR REVISION**

*Re-submission Checklist*

*Published Peer Review*

*PLOS Data Policy*

*Blot and Gel Data Policy*

Please don't hesitate to contact us if you have any questions or comments.

Best wishes,

Ines

--

Ines Alvarez-Garcia, PhD

Senior Editor

PLOS Biology

Carlyle House, Carlyle Road

Cambridge, CB4 3DN

+44 1223–442810

---

## [Decision Letter · Decision Letter 2]

7 Aug 2020

Dear Dr Bucher,

Thank you for submitting your revised Research Article entitled "Sequence heterochrony led to a gain of functionality in an immature stage of the central complex: a fly-beetle insight" for publication in PLOS Biology. I'm handling this manuscript temporarily while my colleague Ines Alvarez-Garcia is out of the office. I have now obtained advice from two of the original reviewers and have discussed their comments with the Academic Editor. 

Based on the reviews, we will probably accept this manuscript for publication, assuming that you will modify the manuscript to address the remaining points raised by reviewer #2. We expect to receive your revised manuscript within two weeks. Your revisions should address the specific points made by each reviewer. In addition to the remaining revisions and before we will be able to formally accept your manuscript and consider it "in press", we also need to ensure that your article conforms to our guidelines. A member of our team will be in touch shortly with a set of requests. As we can't proceed until these requirements are met, your swift response will help prevent delays to publication.

*Copyediting*

*Published Peer Review History*

*Early Version*

*Submitting Your Revision*

Sincerely,

Roli Roberts

Roland G Roberts, PhD

Senior Editor

PLOS Biology

on behalf of

Ines Alvarez-Garcia, PhD,

Senior Editor,

ialvarez-garcia@plos.org,

PLOS Biology

REVIEWERS' COMMENTS:

Reviewer #1:

[identifies himself as Chris Doe]

The authors have done a superb job of addressing my comments. No further experiments or modifications are needed. In my opinion the paper is suitable for publication.

Reviewer #2:

[identifies himself as Stanley Heinze]

The revised version of this manuscript is much improved and, indeed, it was a pleasure to read, with figures as beautiful as in the previous version. All my concerns and issues were addressed to my full satisfaction and the text has been streamlined considerably. The additions to the discussion and the much shorter introduction now present a clear story that stays on message and that should be relevant to many researchers across a range of fields. The study is experimentally very well done and contains a wealth of detailed anatomical and developmental information. It is clearly the culmination of many years of work and, importantly, it opens new avenues of research that are now also explicitly mentioned in the discussion. 

I have a few minor, editorial issues that should be fixed, but that will not preclude publication.

Line 226: please explain what SME means in the figure legend

Line 320: Did the last sentence of the previous paragraph not answer the question posted here? This section investigates this issue in much more detail and more thoroughly, so nothing is wrong with the content. Please rephrase to avoid the apparent contradiction of asking an already answered question. 

Line 442: I don't see how the change in shape corroborates that the CX in larvae is immature but functional. Also, the sentence is quite repetitive, so I suggest to leave out the half sentence after 'corroborating'.

Line 567: Pleas specify the species of bee, I assume honeybee, but maybe be more precise.

Line 694: naming statement regarding CBL and CBU: I think this sentence has become obsolete, as now, EB and FB are used.

---

## [Editor Report · Decision Letter 3]

18 Sep 2020

Dear Dr Bucher,

On behalf of my colleagues and the Academic Editor, Claude Desplan, I am pleased to inform you that we will be delighted to publish your Research Article in PLOS Biology. 

Early Version

PRESS 

Kind regards,

Vita Usova

Publication Assistant, 

PLOS Biology

on behalf of

Ines Alvarez-Garcia,

Senior Editor

PLOS Biology